behaviour/ecology/evolution

alternative food resources, common eiders, foraging performance, optimal foraging theory, polar bears, seabird eggs

**Author for correspondence:**
Patrick M. Jagielski
e-mail: Patrickmjagielski@gmail.com

# Polar bears are inefficient predators of seabird eggs

Patrick M. Jagielski[1], Cody J. Dey[1], H. Grant Gilchrist[2], Evan S. Richardson[3], Oliver P. Love[4] and Christina A. D. Semeniuk[1]

[1]Great Lakes Institute for Environmental Research, University of Windsor, 401 Sunset Avenue, Windsor, ON Canada, N9B 3P4
[2]Science and Technology Branch, Environment and Climate Change Canada, Ottawa, ON Canada
[3]Science and Technology Branch, Environment and Climate Change Canada, Winnipeg, MB Canada
[4]Department of Integrative Biology, University of Windsor, Windsor, ON Canada

PMJ, 0000-0003-4539-7905; CJD, 0000-0003-4947-8972;
HGG, 0000-0001-5031-5092; ESR, 0000-0003-3172-5592;
OPL, 0000-0001-8235-6411; CADS, 0000-0001-5115-9853

Climate-mediated sea-ice loss is disrupting the foraging ecology of polar bears (*Ursus maritimus*) across much of their range. As a result, there have been increased reports of polar bears foraging on seabird eggs across parts of their range. Given that polar bears have evolved to hunt seals on ice, they may not be efficient predators of seabird eggs. We investigated polar bears' foraging performance on common eider (*Somateria mollissima*) eggs on Mitivik Island, Nunavut, Canada to test whether bear decision-making heuristics are consistent with expectations of optimal foraging theory. Using aerial-drones, we recorded multiple foraging bouts over 11 days, and found that as clutches were depleted to completion, bears did not exhibit foraging behaviours matched to resource density. As the season progressed, bears visited fewer nests overall, but marginally increased their visitation to nests that were already empty. Bears did not display different movement modes related to nest density, but became less selective in their choice of clutches to consume. Lastly, bears that capitalized on visual cues of flushing eider hens significantly increased the number of clutches they consumed; however, they did not use this strategy consistently or universally. The foraging behaviours exhibited by polar bears in this study suggest they are inefficient predators of seabird eggs, particularly in the context of matching behaviours to resource density.

## 1. Introduction

Animals make decisions that affect their overall fitness [1–3], and optimal foraging theory (hereafter, 'OFT') suggests that animals

procure prey by following a set of rules to optimize their performance [4,5], enabling them to maximize net energetic returns [6,7]. While animals can at times deviate from expectations of OFT (since optimality can be constrained by various genetic, cognitive, physiological and environmental mechanisms [8–12]), natural selection has nonetheless shaped these behaviours to be adaptive overall within expected environmental contexts [13–15]. However, human-induced rapid environmental change [16] may force consumers to forage on less-preferred prey as their primary prey becomes increasingly more difficult to obtain. Animals foraging on alternative prey may need to adjust their foraging tactics, or risk experiencing a mismatch in cue-response systems resulting in the inability to maximize net energetic benefits [16–18]. The question of whether species can adapt to procuring alternative prey efficiently (and thus maximize net energetic gains), or whether foraging on alternative prey poses long-term negative consequences (or simply reflects a period of adjustment), first requires exploring whether foraging performance, and by extension, foraging decisions, follow predictions of OFT. Indeed, examining the foraging behaviours of animals that increasingly rely on alternative prey stands to improve our understanding of the degree to which foraging performance in anthropogenically altered ecological systems can be described by OFT [19]; an especially important consideration given that foraging performance is linked to fitness and population dynamics.

In the circumpolar Arctic, which is undergoing rapid environmental [20,21] and ecological change [22], polar bears (*Ursus maritimus*) are experiencing behavioural shifts in their foraging ecology in some parts of their range. Although polar bears traditionally feed on seals and other marine mammals hunted on the sea-ice [23,24], they are known to opportunistically forage on land as well, with reports of such behaviours dating back to the fifteenth century [25]. In recent years, however, the inclusion of terrestrial resources has apparently increased for polar bears occupying the southernmost extent of their range [26,27], potentially as a result of missed ice-based hunting opportunities due to changes in sea-ice phenology [21]. Some authors have suggested that terrestrial resources may help offset lost ice-based hunting opportunities [28,29], although this assertion remains unknown [30]. Nevertheless, the increasing length of time polar bears spend on land [31,32], coupled with the extent to which they are witnessed feeding on terrestrial diet items (e.g. [26,27,33–35]), necessitates examining their foraging performance (i.e. decision-making heuristics) as efforts to conserve this species have become an international priority [36]. Understanding polar bears' foraging performance in terrestrial environments is especially important given that terrestrial resources may require different foraging tactics (e.g. climbing cliffs [34]; grazing [35] and chasing prey [37]), and a different suite of environmental stimuli (hereafter 'cues') than what polar bears use to catch seals on the sea-ice [38], potentially affecting their ability to maximize energetic returns. Specifically, reports of polar bears foraging on eggs in avian colonies have increased in recent years as changes in sea-ice phenology have now temporally aligned the onshore arrival of bears to the breeding schedule of birds, in some areas of the Arctic [34,39–43]. While polar bear foraging on seabird eggs has been reported in the past (e.g. [44–47]), these have mainly been incidental occurrences; therefore, polar bears' foraging performance in seabird colonies may be poor given that they have evolved to forage on large marine mammals that occur at low densities [23].

In this study, we examine the fine-scale decision-making behaviours of polar bears while foraging in a large common eider (*Somateria mollissima*) seaduck breeding colony. We use direct behavioural observations of polar bears foraging on seaduck eggs, and apply a descriptive approach guided by classical optimality theory and previous empirical research on foraging behaviour to explore decision-making strategies of polar bears foraging in a terrestrial environment as they deplete the resource. In a previous study in this system, polar bears were observed to consume clutches at a decelerating rate (indicating declining resource density) until they depleted the colony of eggs [48]. Therefore, we examined whether foraging bears are responsive to declining resource density by sampling the area and exhibiting behaviours in accordance with expectations of OFT that would minimize net energetic expenditure. Specifically, by (i) using nest site information, such as the odour of a clutch or conspicuousness of eggs [49], to avoid 'already predated' nests [50] since foraging in an already-searched-area is both time and energetically costly and may have implications on patch-residency time decisions (i.e. marginal value theorem [51,52]). (ii) Adjusting their movement sinuosity in relation to resource density (i.e. area-restricted search theory [53,54]) as a means of limiting energetically costly time spent searching and maximizing nest encounter rates (i.e. slow and sinuous movement in a concentrated area when resources are in high abundance versus faster walking speed and more straight-line movement when resources become less abundant). (iii) Modifying their 'selectivity' in ingesting clutches of eggs of perceived 'lower quality' (i.e. eggs covered in eider faeces, which is the result of the eiders' predatory defence mechanism [55], as the highly alkaline faecal matter may irritate predators) in accordance with resource availability (i.e. foragers are more selective when resources are in high

abundance versus less selective as the resource depletes [56–58]). We also examined (iv) whether using visual cues (i.e. flushing eider hens) aids bears to locate nests [59–61]. We predicted that as the season progressed and resources declined, bears arriving later in the season would demonstrate different behaviours to those arriving early. Specifically, bears would: (i) visit significantly fewer empty nests later in the season even as fewer nests are visited overall; (ii) move with greater sinuosity early in the season when full clutches are more abundant versus more straight-line energy-minimizing movement as clutches deplete; (iii) be more choosey (i.e. ignore more clutches) early in the season when full clutches are abundant. We also predicted that (iv) bears would use visual cues to locate nests throughout the season regardless of resource abundance. Given that terrestrial resources are becoming increasingly used by polar bears [27], our goal was to determine if polar bears foraging on seabird eggs behave in a manner consistent with the expectations of OFT.

# 2. Material and methods

## 2.1. Study site

### 2.1.1. Location and physical characteristics of the study site

We conducted our research on Mitivik Island, a 24-hectare island characterized by low-lying (less than 8 m) tundra interspersed with granite rocks and ponds. The island is situated within Southampton Island's East Bay, in northern Hudson Bay, Nunavut (64°01′47.0″ N, 81°47′16.7″ W; figure 1); a seasonally ice-free region of the Arctic.

### 2.1.2. Eider nest site characteristics and breeding phenology

During the summer months (i.e. June to August), thousands (up to 8000 pairs between 2002 and 2013 [62]; with more recent estimates of 1500–1700 pairs in 2017) of common eiders (*S. m. borealis* spp.) converge onto Mitivik Island and form a dense breeding colony. Birds nest in 'cups' on the ground within the mossy and rocky terrain of the island (figure 2). In 2017 (our study year), eiders arrived to the island approximately in late June (mean arrival (Julian) date: $172.12 \pm 0.22$ days (21 June); median arrival date: $172.25 \pm 0.31$ days (21 June)); and initiated laying a few days later (mean lay date: $175.65 \pm 0.60$ days (25 June); median lay date: $175.71 \pm 0.83$ days (25 June)). Our observations took place on 10–20 July, which is approximately midway through this colony's mean incubation stage [63], so nest replenishment (due to re-laying post-predation) or late lay initiation was probably minimal during this study.

### 2.1.3. Polar bear foraging phenology at Mitivik Island

Every spring and early summer, polar bears migrate northward into East Bay to reach frozen ice in the Foxe Basin before the sea-ice disappears completely. The presence of bears on the island now overlaps with incubating eider hens in June and July; consequently, bears are now increasingly foraging on the eggs of the eiders in this colony [42].

## 2.2. Polar bear observations

We recorded foraging bears with drones (DJI Phantom 3 Pro and 4 Pro models, www.dji.com) between 5.30 and 20.30 h. We initiated filming when conditions were suitable for flying and when bears were actively foraging on eggs. Bears were filmed as soon as researchers noted them on the island and were recorded until they either left the island or were resting for extended periods of time. The drone pilot and observer were stationed on the roof of a research cabin and launched/landed the drone within an electrified fence surrounding the research station. Drones were flown above the focal bear at altitudes that elicited minimal apparent behavioural responses (we suspect that any minimal behavioural response was also due to ambient noise of the colony, and bears continually being harassed by herring gulls (*Larus argentatus*; figure 2a), although we cannot say with absolute certainty that bears and eiders were not affected physiologically (i.e. changes in heart rate [64,65]). Videos were recorded at 30 frames per second at a resolution of $2700 \times 1520$ pixels. In total, we recorded 995 min of polar bear foraging footage. Video data were binned into distinct 'foraging bouts', which represented a near-continuous observation of a bear foraging. Whenever there was a considerable time gap (mean = 167 min; median = 62 min; range = 426 min) in filming a focal animal, due to (i) a

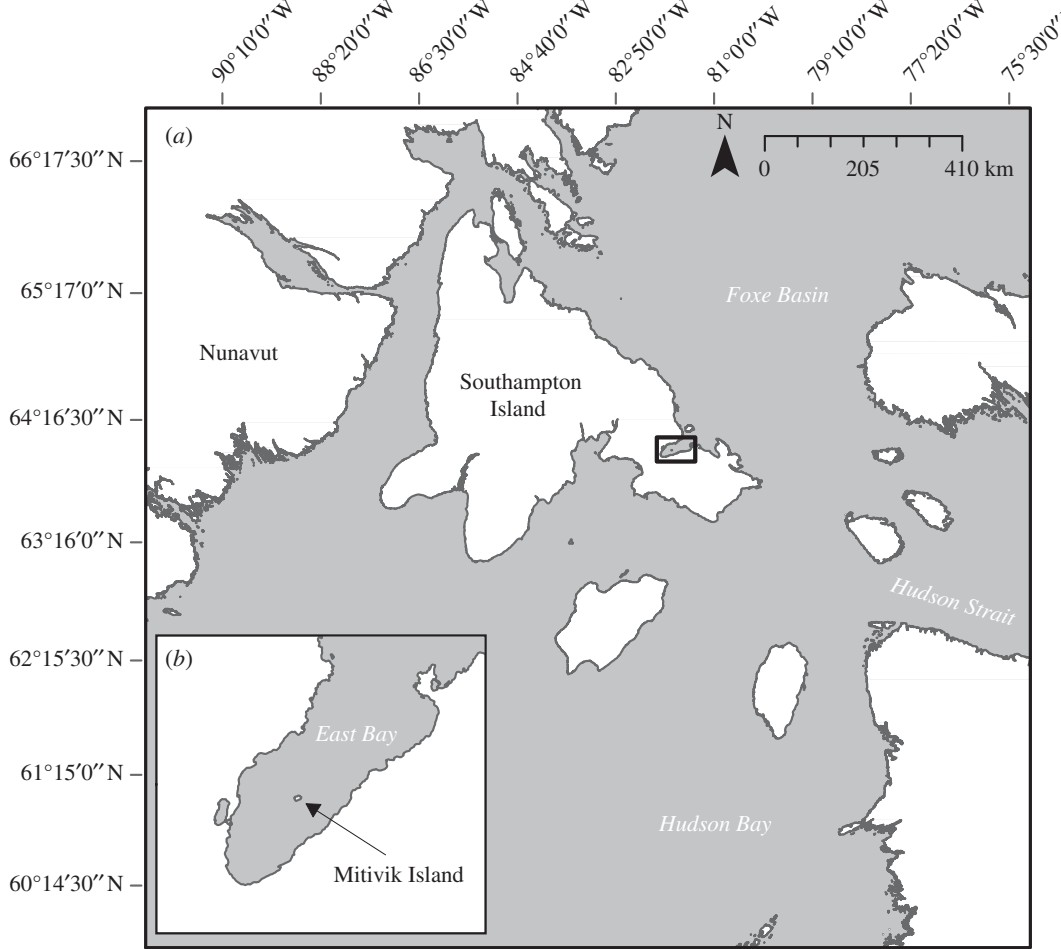

**Figure 1.** Regional location (Southampton Island, northern Hudson Bay) (*a*). Inset: study site (Mitivik Island) in East Bay (*b*).

change in bears' activity (e.g. swimming or resting) which affected the continuity of a foraging bout, and/or (ii) as a result of having to replace batteries due to flight time limitations of the drones used in this study (i.e. approx. 22 min), we considered it a new foraging bout. Importantly, each video was timestamped and foraging bouts were placed in chronological order which served as a proxy for resource depletion (i.e. declining resource density over time) as bears consumed clutches at a decelerating rate until they depleted the colony [48].

Whenever possible, we distinguished individual bears by using: (i) the date and time of video filming, (ii) field logs containing information such as number of bears on the island on a particular day and time, and/or (iii) the stains on bears' fur and scars on bears' faces/body. However, when a bear could not be distinguished from other individuals foraging in the colony during the same time period, we considered it a new individual. In the absence of distinguishing features, bears were also considered 'new' individuals between days. We acknowledge identifying individual bears within and across days was a challenge and strove to find identifying features when possible. However, because we observed multiple bears on a given day (C.J. Dey and E.S. Richardson 2017, personal observation), we can assert any random variance from individual-level effects across days would be swamped by the main effect of the declining resource over time. In addition, while OFT is generally used to describe the resource use of an individual, this framework remains applicable to our study since OFT can act as a guide to describe general foraging behaviours at the population, guild and species levels without the tracking of individual ID [66–68], with such methods also being applied to Ursids [58].

Our dataset consists of a maximum of 20 individual bears with 31 distinct foraging bouts (as some bears foraged more than once in a day) that range from 2.85 to 134 min (mean = 32 min; median = 26 min; range = 131 min). For further details on drone operations, including how we minimized bear disturbance, see the Drone Reporting Protocol [69] in electronic supplementary material.

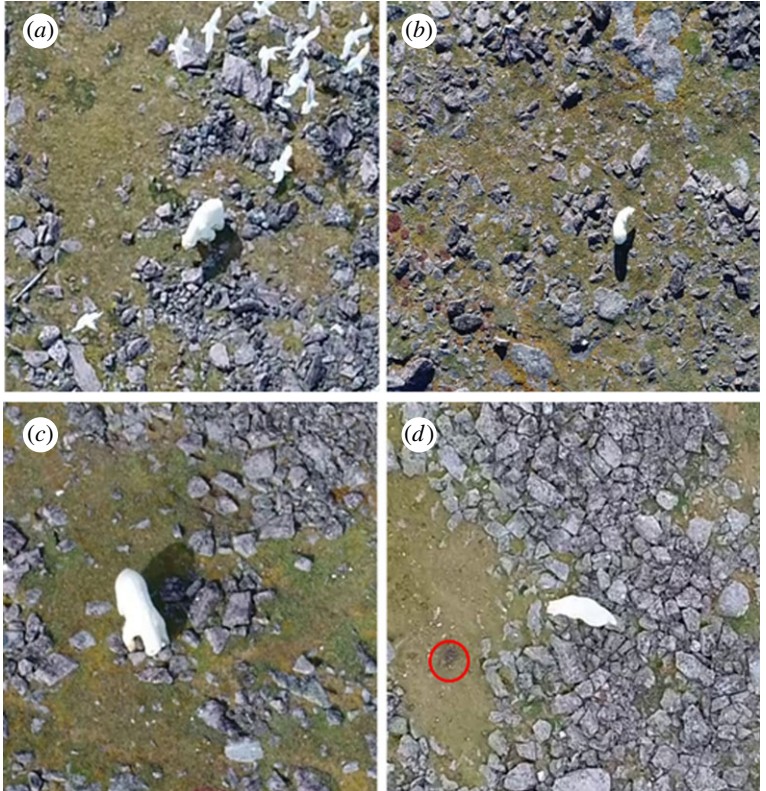

**Figure 2.** Polar bear visiting an empty nest cup (*a*), polar bear initiating a right-hand turn (*b*), polar bear visiting a full clutch (which represents our ability to see a bear either consume or ignore a clutch) (*c*) and polar bear turning in response to a flushing eider hen (in red circle) (*d*). Note: (i) (*a,c,d*) are zoomed in and (ii) nests (full and empty) and eiders were more visible in the live video footage.

## 2.3. Behavioural analyses

We used Solomon Coder version: beta 17.03.22 (https://solomon.andraspeter.com/), a manual behavioural coding tool, to analyse polar bear foraging behaviour. We loaded drone video data into this program and categorized bear behaviour (during video playback) based on predefined behaviours of interest. In reviewing each foraging bout, we recorded: (i) '*Empty nests visited*': number of empty nests a bear was considered to have visited when it was clearly evident from the video footage there were no eggs in the nest cup (see electronic supplementary material, video S1a and figure 2*a*). (ii) '*Movement sinuosity*': number of turns a bear made when searching for nests. We considered it a turn only when it was clearly visible that a bear abruptly (i.e. in one to five bear strides) veered a minimum of 45° from its heading during locomotion or changed directions (i.e. turned left, right, or 180°) after briefly stopping. We did not consider it a turn when a bear gradually (i.e. in more than five strides) moved in another direction (see electronic supplementary material, video S2 and figure 2*b*). (iii) '*Nests ignored*': number of nests a bear was considered to have ignored when it walked up to a full clutch and did not consume any eggs (see electronic supplementary material, video S1b and figure 2*c*). When it was clearly evident that a bear inspected a nest (i.e. looked into the nest, sniffed the clutch) and then ignored it, we concluded that this was due to a bear choosing not to consume the eggs. This may possibly be due to bird faeces on the eggs as a result of the eiders' defence mechanism when flushing from their nests [55]. Therefore, for the purposes of this study, we assumed nests were 'ignored' due to a bear's reduced preference for that particular clutch of eggs. (iv) '*Visual cues ignored and used*': number of visual cues a bear ignored (i.e. eider hen(s) flushing from nest) when it was clear that the focal bear had observed at least one duck flush (i.e. head was facing in the same direction as flushing hen(s)), and did not approach the newly abandoned clutch. Conversely, a bear was considered to have used a visual cue when it was clearly evident that it had observed at least one duck flush (i.e. head was facing in the same direction as flushing hen(s)), then switched its current heading to approach the exposed nest. Ducks flushing in the bears' peripheries were not considered in this analysis (see electronic supplementary material, videos S1c; S3a and

**Table 1.** Statistical models (a–e) used to test drivers of polar bear foraging behaviours on common eider eggs at Mitivik Island, Nunavut. For models (a–d), foraging bout order is the continuous predictor variable (i.e. first recorded foraging bout = 1, last recorded foraging bout = 31) as it serves as a proxy for resource density since bears continually consumed clutches until depleting the colony [48]. For model (e), proportion of cues used (i.e. cues used divided by cues observed by bears) is the predictor variable, since it encompasses the entire suite of events when a cue(s) was present and available for a bear to use. For models (a–e), searching time length (i.e. time spent walking and standing during a foraging bout) is added as a fixed effect to account for differences in filmed-video lengths.

| models | objectives | model variables |
|---|---|---|
| a | contextualize search efficiency | total nest visits ~ resource density + search duration |
| b | searching efficiency | empty nest visits ~ resource density + search duration |
| c | movement efficiency | movement sinuosity ~ resource density + search duration |
| d | selectivity efficiency | nests ignored ~ resource density + search duration |
| e | utility of visual cues | clutches eaten ~ resource density + proportion of cues used + search duration |

figure 2d). (v) 'Clutches eaten': number of clutches a bear was considered to have eaten when it was clearly observable that it was chewing, licking and/or egg contents were dripping when its face was in the nest (see electronic supplementary material, video S3b and figure 2c). In addition, to further confirm an eating event, a full clutch had to be clearly visible upon approach, and/or a hen was seen flushing from the nest when the bear approached. Any approach to a nest not fulfilling the above criteria was considered an 'empty nest visit'. Lastly, we analysed (vi) 'Total number of nests visited': which was the sum of clutches eaten, empty nests visited and nests ignored. This analysis is meant to contextualize the number of empty nests bears visited as the eider breeding season progressed. For example, should the number of empty nests encountered decline while bears increased total number of nests visited, the implications of this behaviour (increased foraging efficiency) would be different should the opposite trend be observed (decreased foraging efficiency). For further detail on behaviours at a nest, see [48]. In addition to these foraging behaviours, we also recorded the duration of time a bear spent walking and standing (i.e. searching) to account for differences in filmed-video lengths in our models.

## 2.4. Statistical analyses

We used generalized linear models (GLMs) and accounted for overdispersion found in the data by using negative binomial error distributions. Each of our models tests different components of foraging efficiency. Using the following response variables (a–e), we tested the following models: *searching efficiency* as (a) the total number of nests visited, and contrasted it with (b) the number of empty nest visits; *movement efficiency* with (c) the number of turns bears made (i.e. movement sinuosity); *selectivity efficiency* using (d) the number of nests ignored; and *utility of visual cues* by analysing (e) number of clutches eaten. For models (a–d), we used foraging bout order as a continuous predictor variable (i.e. first recorded foraging bout = 1, last recorded foraging bout = 31) which serves as a proxy for resource density (since bears' clutch consumption rates decline with time [48] and are, therefore, indicative of a declining resource base). For model (e), we used the proportion of cues used (i.e. cues used divided by total cues observed) as the main predictor variable, which serves to test whether using visual cues enhances bears' ability to locate a greater number of nests. For all our models, we added search duration (i.e. time bear spent walking and standing during a foraging bout) as a covariate to account for differences in filmed-video lengths. See table 1 for model overviews.

All statistical analyses were performed in R v. 3.4.4 [70] using the tidyverse [71] and glmmTMB [72] packages. Figures were created using the ggplot2 [73], ggeffect [74], cowplot [75] and gridExtra [76] packages.

# 3. Results

As the eider-duck breeding season progressed, the total number of nests that bears visited declined significantly ($p < 0.001$; figure 3a and table 2), with bears marginally (i.e. significant at the 0.1 level) increasing their visits to empty nests during the same period ($p = 0.08$; figure 3b and table 2). There was no significant change in movement sinuosity as the season progressed ($p = 0.82$; figure 4a and

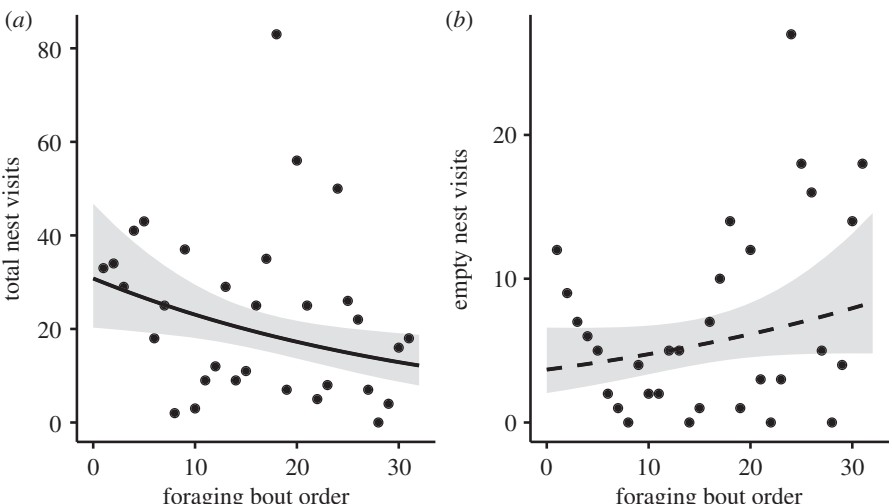

**Figure 3.** Total number of nests visited declined as the eider breeding season progressed (*a*). Empty nest visits marginally increased as the eider breeding season progressed (*b*). Each data point represents a single foraging bout. Grey shading represents the confidence intervals around the predicted means.

**Table 2.** Results of statistical models (a–e): variables, parameter estimates, standard errors (s.e.), *Z* scores and *p*-values representing empirically tested behaviours of polar bears foraging on common eider eggs. *Significant effect (*p* < 0.05). **Tendency towards significance (0.05 < *p* < 0.10).

| models | variables | estimates | s.e. | *Z* | *p*-value |
|---|---|---|---|---|---|
| a | intercept | 2.687 | 0.249 | 10.754 | < 0.001* |
| (total nests visited) | resource density | −0.028 | 0.012 | −2.355 | 0.018* |
| | search duration | 0.000 | 0.000 | 6.541 | <0.001* |
| b | intercept | 0.430 | 0.339 | 1.269 | 0.204 |
| (empty nests visited) | resource density | 0.025 | 0.014 | 1.732 | 0.083** |
| | search duration | 0.000 | 0.000 | 4.681 | <0.001* |
| c | intercept | 2.555 | 0.216 | 11.792 | <0.001* |
| (movement sinuosity) | resource density | 0.002 | 0.009 | 0.228 | 0.82 |
| | search duration | 0.000 | 0.000 | 7.488 | <0.001* |
| d | intercept | 1.046 | 0.497 | 2.103 | 0.035* |
| (nests ignored) | resource density | −0.099 | 0.023 | −4.185 | <0.001* |
| | search duration | 0.000 | 0.000 | 2.378 | 0.017* |
| e | intercept | 1.969 | 0.349 | 5.634 | <0.001* |
| (clutches eaten) | resource density | −0.063 | 0.021 | −2.953 | <0.01* |
| | proportion of cues used | 1.412 | 0.621 | 2.272 | 0.023* |
| | search duration | 0.000 | 0.000 | 5.233 | <0.001* |

table 2). Bears ignored significantly fewer nests as the season progressed (*p* < 0.001; figure 4*b* and table 2); and the number of clutches that bears consumed increased significantly when a greater proportion of visual cues were used (*p* = 0.023; figure 5 and table 2).

## 4. Discussion

Changing sea-ice conditions have created a phenological overlap on Mitivik Island so that polar bears now move onto land during the egg laying and incubation period of eider ducks. Consequently, bears

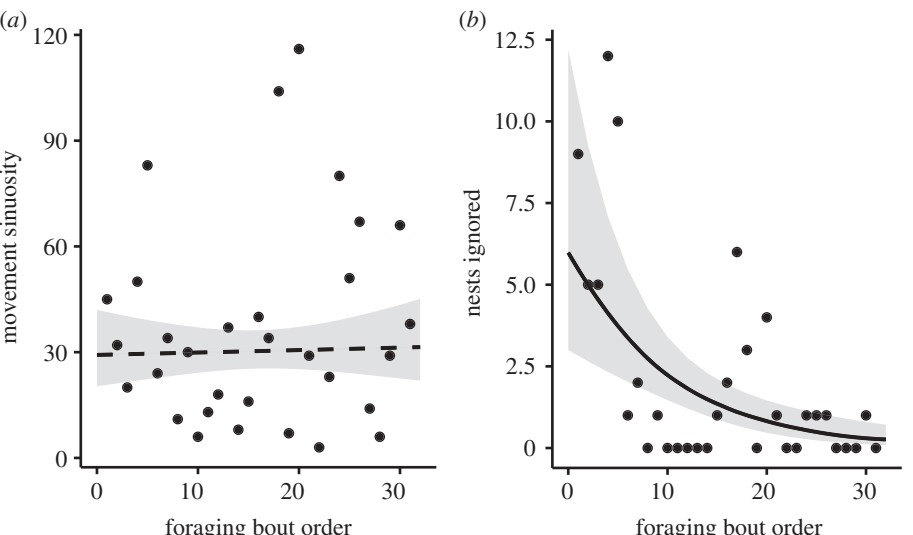

**Figure 4.** Movement sinuosity stayed constant as the eider breeding season progressed (*a*). Nests ignored declined as the eider breeding season progressed (*b*). Each data point represents a single foraging bout. Grey shading represents the confidence intervals around the predicted means.

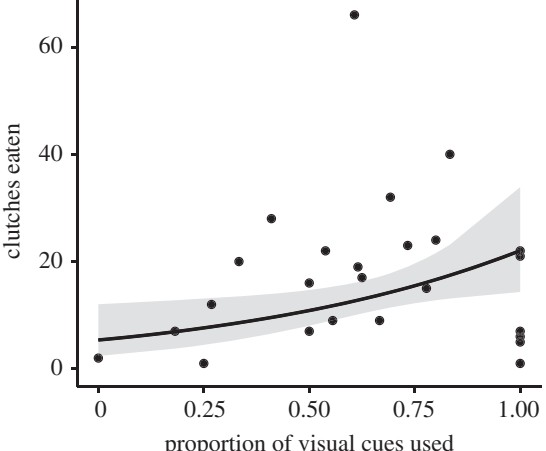

**Figure 5.** Clutches eaten increased with an increase in the proportion of visual cues used (i.e. total cues detected and used to locate nests). Each data point represents a single foraging bout. Grey shading represents the confidence intervals around the predicted means.

are increasingly foraging on eider eggs in this large seabird breeding colony [42], presumably since they are able to accrue important calories when the colony is full of unconsumed clutches [48]. We acknowledge that our sample size is small (i.e. based on one season of observations) and that our results need to be taken in the context of island size and eider nest density. However, based on this present study, our results suggest that polar bear foraging performance does not align with expectations of OFT, in that bears do not behave in accordance to resource density and as a result, do not exhibit evidence of foraging efficiency.

## 4.1. Polar bears do not discern full from already-predated nests

As the eider breeding season advanced, total visits of bears to nests (i.e. clutches eaten, empty nests and nests ignored) declined (figure 3*a*). Simultaneously, bears showed a marginal increase in their visits to empty nests (contrary to predictions; figures 2*a* and 3*b*), suggesting that the proportion of 'nest visits' are increasingly empty, which may indicate that bears are unable to discern full from already-predated nests in advance of visiting them [50], either by using sensory mechanisms such as the odour of a clutch or conspicuousness of eggs [49]. Given that visiting empty nests can be both time and energetically

costly (i.e. by visiting already searched grounds), the seeming inability to recognize which nests are empty before visiting them may have important implications on optimal giving-up times/densities [51,52] in that bears may overstay in the colony. While the energetic cost of foraging in the Mitivik Island eider colony is low [48] and probably has minimal fitness implications as bears walk slowly across a flat terrain and dense colony, we posit that an optimal giving-up time would become particularly critical for bears foraging either late in the eider breeding season when few full nests are left, or in a larger area with fewer nesting hens. Quantifying an optimal giving-up time of polar bears foraging on eggs at a larger spatio-temporal extent would aid in determining polar bears' overall ability to make adaptive patch-residency time decisions (i.e. marginal value theorem [51,52]). Although this is well beyond the scope of the present study, our findings present the first step into a more thorough investigation of long-term fitness consequences of terrestrial foraging for polar bear populations.

## 4.2. Polar bear movement patterns appear suboptimal as they deplete the resource

Polar bears did not exhibit optimal movement patterns in relation to resource abundance as there was no significant relationship found between the number of turns bears made and foraging bout order as the summer progressed and eggs were predated by bears (figures 2b and 4a). While it was not possible to accurately assess bear locomotion speed, our results of polar bear turn rates are nonetheless inconsistent with what would be expected from OFT (i.e. area-restricted searching [53]), in which animals should adjust their time spent in respective search modes based on changing resource quality and abundance. For example, elk (Cervus elaphus) displayed higher movement sinuosity when foraging in high-quality vegetation patches, and transitioned to more straight-line movements when foraging in lower-quality ones [54]. Polar bears foraging in avian colonies appear to forgo the consumption of many nests in their direct foraging path, and instead walk by or ignore nests ([40]; figure 4b). Given that polar bears have a highly developed sense of smell capable of locating seals hiding in their subnivean lairs [77] and that eider hens are abandoning their nests as bears walk through the colony, we would expect polar bears to employ optimal movement modes relative to resource density. Indeed, because common eiders nest in a clumped distribution [78], early in the season when nests were in high abundance bears should have concentrated their time in a restricted area after locating a nest to increase their search efficiency, and then transition to opportunistic feeding as nests depleted to optimize their time and energy cost [53]. They did not do so. We suspect that consistent movement patterns across bears, despite a decline in resource abundance may result from: (i) their inexperience foraging in areas of high prey density in a seabird colony given that their typical marine prey occurs at low densities and are encountered infrequently [38], and/or (ii) satiation from recent feeding in the marine environment prior to coming onto the island as part of their foraging-based migration (and therefore no need to employ a search mode). However, we observed bears actively searching for nests on the island late in the season despite that few clutches remained, making satiation an unlikely universal explanation.

## 4.3. Polar bear selectivity in clutches appears to be optimal

Early in the season, polar bears displayed a higher degree of selectivity in the eggs they consumed and then relaxed their apparent 'choosiness' as the colony was depleted. This was evidenced by the declining trend in the number of nests ignored (figures 2c and 4b). Selective foraging has also been documented among both grizzly (Ursus arctos) and black (Ursus americanus) bears when consuming migratory salmon (Oncorhynchus spp.). When salmon were in high abundance, both species displayed selectivity towards higher quality fish (i.e. those that had not yet spawned), but became less selective and transitioned to spawned fish as the availability of salmon declined [57,58]. The observation that polar bears may be 'choosy' when the colony was full may suggest bears were not in any energetic deficits, perhaps as a result of satiation from having recently come off the sea-ice after the spring hyperphagic period. While selectivity in prey quality is expected to be optimal under natural conditions [56], under human-induced rapid environmental change, such selectivity may prove costly as preferred prey availability and profitability become uncertain and unpredictable [79].

## 4.4. Use of visual cues help polar bears locate more nests

The use of visual cues of flushing eider hens significantly influenced the number of clutches bears located and consumed (figures 2d and 5), suggesting there is a benefit to learning and adopting this strategy.

Indeed, under experimental settings, both ground squirrels (*Spermophilus beldingi*) and domestic cattle (*Bovidae* spp.) significantly increased their probability of locating a food item when aided with 'novel' visual cues (i.e. 'new plant' and 'traffic barricades and cones', respectively) [59,60]. Similarly, black bears foraging on salmon significantly increased their probability of locating fish when they used visual cues (i.e. salmon carcass) during the daytime [61]. Thus, capitalizing on the use of flushing eider hens as cues should be a strategy adopted continually once learned. Interestingly though, we observed that polar bears did not universally nor consistently (i.e. within a foraging bout) use this strategy in the current study. We postulate that such underutilization may emerge as a result of non-mutually exclusive combination of the following: (i) bears may not have had enough opportunities foraging in an avian colony to learn this strategy, (ii) flushing eider hens do not resemble cues bears typically experience so they are not perceived as valuable (i.e. 'undervalued resource' [80]), and/or (iii) bears were satiated (again, possibly from recently feeding on the ice) with no need to actively seek out cued nests. While this study demonstrates the utility of visual cues for bears to locate concealed nests, it is worthwhile to mention that flushing eider hens do not always indicate a nest (e.g. when an eider is not sitting on its nest but flushes nonetheless as the bear approaches, or when an eider uses a broken-wing display to draw the bear away from its nest). Therefore, there remains a need to estimate the reliability of this cue, which, while beyond the scope of the current study, will help to approximate the absolute value of visual cues to foraging bears.

## 4.5. Polar bears may learn to improve their foraging performance over time

While our results suggest that polar bears do not necessarily adhere to all expectations of OFT, this is not surprising given that lengthier terrestrial residency has only recently become an increasing occurrence for polar bears at a population level [31,32]. Variation in foraging efficiency may be due to differences in bears' experience levels, and polar bear terrestrial foraging behaviours may simply reflect a period of reinforced learning [81,82]. We, therefore, postulate that polar bears can improve their foraging performance should they continually be driven onto near-shore terrestrial environments and overlap with the incubation period of ground-nesting birds (as seen here: [34,39–43]). Given that polar bears are capable of learning to use resources that have proven to be profitable in the past [83], individuals who have experienced foraging success in avian colonies, such as those reported consuming hundreds of waterfowl eggs over a few days [39,84], may be driven via memory to forage on eggs over several years and, as a result, become more adept at the task.

## 4.6. Long-term profitability and sustainability of this predator–prey interaction are unknown

Recent years have seen an increase in reports of polar bears foraging in eider colonies [42,43,84]. However, most of these reports (including this study) note only a handful of bears in the colonies and, even with over 300 000 breeding common eider hens inhabiting the Canadian Arctic [85], only approximately 30% of colonies are visited by bears on a regional scale [42]. Polar bears also deplete the resource quickly and only a few individuals gain appreciable calories [48]. Combined, these lines of evidence suggest eider eggs are not yet a significant resource for polar bears at a population level [30], so while efficient foraging behaviour may impact the number of calories an individual bear accrues, the overall importance of eggs as a resource for polar bears will remain low unless bears' use of available colonies intensifies. However, should the number of bears foraging in eider colonies increase, the consequences for eider populations may be devastating [86,87], making this resource unsustainable in the long run (but see [88] where predictive modelling suggests otherwise under certain conditions).

# 5. Conclusion

Our findings here suggest that polar bears are inefficient predators of seabird eggs, particularly in the context of matching foraging behaviours to resource density. If polar bears are limited in their ability to accurately assess the quality of their foraging patch and do not adjust their behaviours accordingly, the energy individual bears gain from eggs may be less than previously estimated [48]. These results may, therefore, influence the previously predicted energetic consequences of this climate-mediated behavioural shift at a population level [89]. We acknowledge that our sample size is small, and therefore suggest that a long-term spatial and temporal study is required to better understand not only the long-term fitness consequences of polar bear foraging performance beyond Mitivik Island,

but also the ecological implications of this predator–prey interaction. This study demonstrates that, while species are able to incorporate 'less preferred' resources into their diet when their primary prey becomes more difficult to obtain, they may not be able to do so efficiently.

Ethics. This research was approved by the Environment and Climate Change Canada Animal Care committee (no. EC-PN-17-026), the Nunavut Research Institute (via Nunavut Wildlife Research Permit #WL 2017-030) and the Aiviit (Coral Harbour) Hunters and Trappers Association.
Data accessibility. Data available from the Dryad Digital Repository: https://doi.org/10.5061/dryad.prr4xgxjw [90].
Authors' contributions. C.A.D.S, C.J.D., E.S.R., H.G.G. and P.M.J. conceived the study; C.J.D. and E.S.R. conducted the fieldwork; P.M.J. analysed the data; P.M.J. and C.A.D.S. ran statistical analyses; P.M.J. wrote the manuscript; C.J.D., E.S.R., H.G.G., O.P.L. and C.A.D.S. provided comments and suggestions to the manuscript; C.A.D.S., C.J.D., E.S.R., O.P.L. and H.G.G. obtained funding.
Competing interests. We have no competing interests.
Funding. This work was supported by Environment and Climate Change Canada; Mitacs (grant no. IT04216); Baffinland Iron Mines; Natural Sciences and Engineering Council (grant no. 06768) and The Liber Ero Foundation.
Acknowledgements. We thank Mike Janssen, Jake Russell-Mercier, Holly Hennin and Bronwyn Harkness of Environment and Climate Change Canada (ECCC) for their logistical support, and Josiah Nakoolak, Jupie Angootealuk, Clifford Natakok and Bob Hansen for their assistance with fieldwork.

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
