## [Peer Review File · Royal Society Open Science]

Review History

RSOS-201733.R0 (Original submission)

Review form: Reviewer 1

Is the manuscript scientifically sound in its present form?

Yes

Are the interpretations and conclusions justified by the results?

No

Is the language acceptable?

Yes

Do you have any ethical concerns with this paper?

No

Have you any concerns about statistical analyses in this paper?

No

Recommendation?

Major revision is needed (please make suggestions in comments)

Comments to the Author(s)

This study used drones to examine the foraging behavior of polar bears at common eider nests on Mitvik Island during a 10 day period in 2017. The authors focused on determining whether behavior follows patterns expected per optimal foraging theory. The authors provide some interesting and informative results about polar bear behavior at seabird colonies. This is particularly important because direct observational studies of polar bear foraging are overall quite rare, including of their more traditional prey on the sea ice. Thus, despite the small sample size and short duration of the study, the data are important to publish. That said, more caution seems warranted in generalizing the potential results. At most the data are from 20 bears and we don't know how few that might be (3, 5, 10?). Thus, I'd suggest that the authors be somewhat more cautious in the conclusions drawn. I struggled to link the analyses to the objectives. I had to read it several times to make sense of what was done and why, and didn't fully understand what some of the analyses were getting at until I read the discussion (i.e. the models were clear but the rationale was not). More background is needed to support that the foraging bout order is an appropriate proxy for resource abundance (see specific comments below) since that is a key aspect of the paper (and the terms need to be used consistently - foraging bout versus event, etc.). Importantly, I think more caution is warranted when discussing the context of polar bear use of bird eggs broadly (see my specific comments below) as it is overstated in several places. The conclusion of the paper is that polar bears are not yet efficient predators of seabird eggs, yet I didn't find the evidence for that conclusion convincing, in part because of the low sample size and short duration of the study but also based on the metrics examined (i.e. frequency of turning and selectivity). I would be more convinced that their "behavior" wasn't optimized which is different than saying they aren't efficient predators - the latter really requires knowing whether they balanced energetic gain with the expense of foraging. I think that some of the most important findings here are not highlighted. For example, it is significant that you documented a decline in nest visitation and increase in encountering of empty nests in just a 10 day period by less than 20 bears. This suggests that it is a quickly depleted resource that is not likely to sustain bears for any appreciable amount of time unless within that 10 days they are packing on extra pounds and there are similar sites throughout this region? - not to mention long-term sustainability of the food resources as a result of potential decline in eider populations via reduced reproduction. Is Mitvik an anomaly or are there similar seabird nesting throughout this region that other polar bears may be using (i.e., can some context be given for whether this site is similar to many others in the region or is somewhat rare?). Discussing these topics would make this a greater and more significant contribution than it is currently. I look forward to seeing a revised, published version.

Specific comments

Abstract: Lines 15-17: The statement "In response [to climate mediated sea-ice loss] polar bears are increasingly supplementing ice-based seal-hunting opportunities with eggs from seabird colonies". It's important to note that although there have been reports of polar bears feeding on eggs from seabird colonies (both in the past and in recent years), that few show this increasing over time and associated with sea ice loss (other than that they overlap more because bears come onshore earlier). Overall, this is not a common behavior by polar bears (i.e. there are not a lot of bears exhibiting this behavior even if some bears in a number of locations exhibit this behavior). The vast majority of bears that come to land in the summer throughout the polar bears range either rest and wait for the sea ice to return or scavenge on marine mammal carcasses. Thus, this statement is overgeneralized and should be edited to better reflect the context of observed consumption of seabird eggs by polar bears.

Line 34: It's not clear why the word "continually" is used in this sentence.

Line 72: This sentence sets up the expectation that you are measuring intake rates. I'd suggest rewording this section to indicate that you examined whether bears are responsive to declining resource density (which was assumed to occur over time during the observations) by modifying their selectivity, visiting fewer empty nests, etc.

Line 74: Can you be more specific for #2. How might it be expected that they would adjust their movement behaviors in relation to resource density?

Line 75: What would lower quality items be if all the of the resources are eggs? Eggs at various stages and nutritional content?

More detail needs to be provided on the expected directionality and rationale for the relationships examined (i.e. that you would expect them to become less selective as the resource is depleted, etc.)

Materials & methods

Study site: It would be helpful to reference Fig. 2 here and further describe the habitat used by common eiders (i.e. they are nesting in ground nests among vegetation and rocky areas, etc.).

Information on the timeframe in which they create nests and lay eggs as well as how synchronous egg-laying is are important context for the study.

Line 113: Is a foraging event the same as a foraging bout?

Line 117-119: Using this information it would be helpful to have some idea of the possible minimum number of bears (i.e. could this be repeated data from 3 bears or is it more likely 10 or more?). This is important because the manuscript is based on a very temporally constrained data set (i.e. 10 days of data in a single year).

Line 122-124: This isn't entirely convincing since experience would likely affect questions #1 and #4 at a minimum, if not all of the questions identified in the study (which you acknowledge later in the discussion). Perhaps here you could at least note potential effects of individual experience (either within the year of your study or between years) on behavior.

Line 140-144: It is really interesting that this behavior even occurs. I would have assumed they would eat eggs in any nest. Additional background on bear behavior at the nests would be helpful when you present the objectives.

Lines 157-158: "This analysis is meant to contextualize as the season progresses..." Here you need to go ahead and explain how you added the temporal component in the analysis. You explain it in the statistical analyses, but I was left wondering about how that was being incorporated and didn't get the answer until the statistical analyses section. Also, it's helpful earlier on to understand that you used foraging bout/event

Line 170-172: I think I've answered my question here that foraging event is the same as foraging bout? Table 1 indicated "foraging event". I'm reviewing the manuscript assuming these are the same thing. Obviously, if that is the case, the same term needs to be used throughout. Also, in line 172, perhaps clarify that foraging bout/event is a proxy for trends or changing resource density in addition to indicating that it was used as a continuous variable.

Statistical analyses:

Lines 167-172: I struggled to relate the models to the objectives. It would be easier to understand the analysis if Table 1 included a column indicating the question/objective being addressed with each model and had a separate column for the response/dependent variable and independent model variables. The table heading needs to indicate what the foraging event, search time and number of cues used are including that they are continuous variables, even though you have stated that information in the text. This allows the table to stand alone to help the reader understand the variety of models.

At this stage in the paper, I'm still unclear how the models address the questions of "were bears able to limit search time and maximize intake by...." stated in the introduction. I understand that temporal change is assumed to be a result of resource density. But I'm not sure what the expectation is relative to movement sinuosity or selectivity over time (with foraging event). Should they become more selective and move less b/c the return is less? Similarly what is the expectation for movement sinuosity relative to search time? Should they use visual cues more as foraging event increases (i.e. resource density increases). In the models, I'd suggest using a different term than foraging event that you define that would make the models more intuitively related to the questions (i.e. perhaps you use "resource density" as the model parameter but state in the text and in tables & figures that the a proxy of chronological foraging event was used).

It's also difficult for the reader to understand and assess how well foraging event might serve as a proxy for resource density. This seems like it would be the case if all bears are foraging in the same locations such that the order of bouts would definitely affect resource density. But this is unclear from the background provided. Further, how synchronous is egg-laying? Does that play a role in the chronological sequence of resource density?

Results: Line 196 "as predicted..". I don't remember seeing predictions. As stated previously, it has been unclear up to this point what was expected and why some of the relationships were examined.

Line 200-203: I'm not following the logic here. If they exhibited a trend of increasing visits to empty nests how does that demonstrate the ability to discern when nests are empty? Visiting fewer nests overall makes sense. Also, the relationship with visiting empty nests was significant at $p = 0.08$. You should be specific any time you discuss a result that has a $p > 0.05$ as most readers assume that relationships identified in a discussion are significant at $p \leq 0.05$

Figs. 3 and 4 use the term "foraging bout order" - similar comment as previously, I understand this to be the same as a foraging event? Perhaps use "foraging event order" throughout?

Lines 221-223: There is so little background provided on why the number of turns is indicative of optimal foraging that it's impossible to evaluate this finding. In the introduction the expected relationship needs to be described (which I presume is more sinuosity when there are more nests). The information in lines 233-236 should be provided in the introduction (i.e. this was the hypothesis you were testing).

Lines 236-237: I again question the potential effects of the assumption that foraging event/bout/bout order (which I understand to all be the same measure) is perfectly measuring resource abundance. Either you need to make a stronger case for that or acknowledge the potential that your proxy is imperfect and could have affected the lack of an apparent relationship with sinuosity.

Lines 245-257: Any thoughts on the basis for selectivity? What is known about nutritional quality of the eggs as they incubate? Do other predators avoid eggs with fecal matter as you suggest? If you have no idea what might cause selectivity, even saying that would be informative. It is surprising to me that they are selective.

Line 280: Perhaps reword so "not necessarily" is not included twice in the same sentence.

Line 300-302: This statement is not convincing. Assessing whether polar bears can "adapt to a lengthier terrestrial residency" is not fully addressed by determining whether they use foraging strategies similarly in terrestrial and sea ice environments and/or exhibit behavior consistent with optimal foraging theory. Ultimately it depends on the quality and abundance of food (and their intake of it) in relation to the energetic cost to obtain it.

Line 302-304: Your results don't provide any information about maximum net energy gain. The conclusions you can make are only about behaviors relative to optimal foraging theory.

Line 307: satiated from what? Having just come off the ice and feeding on seals? If so, is there evidence of recent feeding (i.e., defecating) that might indicate whether bears are feeding just prior to come to the island?

Line 308-310: This seems pretty speculative given the data (i.e. collected over a 10 day period in a single year). In the next sentence you acknowledge the small sample size, but I'd suggest not trying to link the short-term behavioral observations in this study to potential implications for body condition.

Review form: Reviewer 2

Is the manuscript scientifically sound in its present form?

Yes

Are the interpretations and conclusions justified by the results?

No

Is the language acceptable?

Yes

Do you have any ethical concerns with this paper?

No

Have you any concerns about statistical analyses in this paper?

Yes

Recommendation?

Major revision is needed (please make suggestions in comments)

Comments to the Author(s)

This is an interesting study on the foraging ecology of an obligate carnivore on an unusual prey item. The sample size is small and I urge the authors greater caution in interpretation. The current manuscript is overly speculative and a more tentative approach would be helpful. Nonetheless, it is a novel study and quite a challenging data set to obtain.

I urge greater caution in presentation of what is known about the foraging ecology of polar bears and its importance. There is heavy reliance on works by Gormezano et al. yet those studies are similar in their findings to Russell (1975 – Arctic 28: 117-129). It is unclear that terrestrial resources use is increasing or that it has any significant energetic value (see Rode et al. 2015).

I have concerns about the conclusion of non-optimal foraging. I think, correct me if I'm incorrect, but you have a staggered entry design for bears. The "trial", in an optimal foraging sense, restarts for every new individual. Because you are not following an individual through time and optimal foraging is an individual response, I don't see the matching between theory and the study design. I'm open to being convinced otherwise.

The paper is well written. I have some specific issues but with some careful reconsideration, the manuscript will make an interesting contribution to the literature.

16 – “increasingly supplementing ice-based seal-hunting” - there is limited support for this statement. I agree that it is increasingly reported but I suspect that this is related, at least in part, to the amount of activity in the Arctic and interest in polar bears. Some of the very earliest reports of polar bears from explorers record eating of eggs (e.g., Samuel Hearne, John Rae). Possibly “increased reports of supplemental feeding” or something similar.

44 – the statement “poses any long-term negative consequences to species impacted” precludes any positive consequences. This seems like an odd statement and somewhat biased.

53-54 – “increasing the bears’ reliance on alternative prey” - you cite a book and 3 papers by a single group of authors. The idea that there is increasing reliance on alternative prey, however, Rode KD, Robbins CT, Nelson L, Amstrup SC. 2015. Can polar bears use terrestrial foods to offset lost ice-based hunting opportunities? *Frontiers in Ecology and the Environment* 13:138-145.

This paper states “Only small numbers of polar bears have been documented consuming terrestrial foods even in modest quantities” and “Where consumption of terrestrial foods has been documented, polar bear body condition and survival rates have declined even as land use

has increased. Thus far, observed consumption of terrestrial food by polar bears has been insufficient to offset lost ice-based hunting opportunities”

A better reflection of the state of knowledge on terrestrial feeding by polar bears is warranted.

57 – a peer reviewed source would be preferable to a report

57 – please clarify what is meant by “This” – is it referring to “conserve” or “foraging performance”

59 – I agree that the cues are different but there is clear evidence in the literature that foraging on birds is not a new behavior and goes far beyond the noted references.

65 – I find the argument that polar bears may have poor foraging performance a bit odd. Polar bears evolved from a brown bear ancestry (fairly recently) and brown bears are omnivores with a very diverse diet. As an olfactory predatory, why should polar bears have poor foraging performance? Polar bears are known to consume a wide variety of species despite their specialization so postulating good performance seems as likely as it being poor. Authors choice on this but it strikes me that a stronger approach is that seaduck eggs may be a novel forage item and thus the situation is akin to prey switching in functional response context.

79-81 – “To our knowledge, this is the first study...” I’m not a fan of such justification – the science should speak for itself. I suggest removing this as it add little.

96-113 – the study is based on 10 days of sampling in a single year, 995 minutes of sampling, a maximum of 20 bears, and 31 foraging events. Given the challenges in collecting the data, this is a small but reasonable sample size. However, it does restrict the ability to generalize.

121 – “new individual” – I have a sense that there is a likelihood of pseudoreplication and that some animals could have been counted repeatedly as being new. This further restricts the works ability to generalize.

124 – optimal foraging theory is an interesting context for the study but it is unclear how patches may be assessed in this context. A polar bear may view the whole island as a patch and have limited ability to move. Within the context of an ideal free distribution, the bears may not be free to move (i.e., swim to a new patch) and know of alternatives. The interactions with other bears may not be free. Because polar bears foraging on terrestrial prey lose mass, I’m not convinced that optimal foraging is the driving force / context. I’m OK with it but do the bears have a choice to move? Is the marginal value theorem operating only at the scale of the island and not the broader landscape? Will a depleted patch (the island) result in a bear moving to a new patch?

142 – I think the argument that feces covered eggs are not consumed? This seems very odd – polar bears are notorious for eating decaying carrion. That duck feces is a deterrent seems a bit odd to me. I’ve not seen evidence of similar behaviour in other egg eating mammals.

Methods – there is no information on how the nests are located / mapped. A brief section on this would be useful. I did not see this in the supplementary files.

172 – watching the videos, I imagine the depletion of nests is also due to gulls? Arctic foxes too?

184-5 – “with bears marginally increasing their visits to empty nests” – this is where low statistical power is hampering analyses. 1 year of data with only 31 observations. There’s no solution but it warrants a caveat rich discussion.

I don't see how individual was assessed in the analyses. I assume some of the observations are repeated for some individuals and others have a single sample? If this is a cross-section of animals, the implications / conclusions may be quite different. For example, a bear on its 10 foraging trip may behave quite differently from a bear on its first trip. I don't see a path to addressing this but again, caution in interpretation is warranted.

193-195 - I wasn't a fan of "first" in the introduction but getting a double dose is even less appealing. Being first is not a meaningful reason for publication.

195 - I don't see how a bear consuming an egg cannot gain energy. Reword / clarify.

196-197 - I have some concerns about the conclusion (and thus the title) because you are not following individuals over time. This is a cross-sectional study and you presumably have a staggered entry into the "trial". I don't see a means of getting past this but assuming non-optimal foraging appears to be a major overreaching of the data / design.

201-207 - yes, animals can detect food. This is in essence the argument being made. A horse can see that there is no grass to be eaten. A crow will scavenge from a carcass it can see. I would reduce this emphasis / condense. It's not an overly novel finding.

214 - the genus is now *Pusa* for *Phoca hispida*

214 - no information is provided on ice conditions. To this point, I did not understand that the bears had the choice to hunt for seals on the ice vs. be on land. More detail on sea ice conditions is needed. Do you have information on how long bears stayed on the island? Did some leave? If the context is seaduck eggs vs. seals, this has to be made clear. If it is this case, eating eggs may yield benefits beyond energy (e.g., micronutrients, vitamins).

229 - "foraging in avian colonies appear to forego the consumption of many nests in their direct foraging path" - to me, this observation suggests that eggs are not a meaningful source of energy. If this is true, the idea of optimal foraging is not meaningful. A core tenant of optimal foraging is that the animal must be foraging. This suggests that the animal is not feeding on available resources, thus not depleting them, and thus, unlikely to forage optimally.

238 - "satiation" - given that a polar bear can consume ca. 10-20% of its body mass and a bear could weigh 500 kg. Conservatively, satiation might be 50 kg of eggs and at 150 grams/egg (based on a goose egg mass), satiation might come with 333 eggs (plus / minus). If there are 1600 pairs with a mean clutch of 4 eggs (possibly high), this means there are ca. 6400 eggs available pre-predation. Does it make sense to invoke a bear eating 5% of the eggs (333/6400) in a single bout? With 31 bouts observed, does this make sense? Invoking satiation is OK but there's no information on the bears provided. If the bears are small (say 200 kg), satiation may make more sense. Your citation on lines 287-291 suggest that my calculations are conservative. The alternate hypothesis, not proposed, is that they're not that motivated / interested in feeding. It is noteworthy that fasting ursids often do not feed even when presented with food. Did 100% of the bears observed feed? It is odd that fasting physiology was not explored as an aspect of a bear's motivation to forage (or not). Perhaps the authors are implying satiation is the equivalent of fasting?

255-257 - delete - this is akin to "do more research" and add little

283-286 - "we therefore postulate polar bears can potentially learn to optimize (and potentially specialize [77, 78]) their foraging performance should they continually be driven onto nearshore

terrestrial environments” This is a bit odd given the analyses of Rode et al. (op. cit.) that suggests terrestrial resources are not a meaningful contribution. If eider eggs were a meaningful energy source, should bears exploit them already? In an ideal free context, shouldn't they give up hunting seals to forage on eggs even if sea ice is present? Maybe my logic is off but if the eggs are a meaningful source of energy, they should be used.

Another caveat, I suspect that heavy predation on an eider colony would reduce their reuse, predictability for the bears, and cause a decline in eiders using the area. The longer term dynamics seem tenuous.

For this section, I urge less speculation.

310 - the caveat of a small sample size shows up very late in the discussion. Move to the start. Readers need a reminder of this from the outset.

296-314 - this is largely a summary / review of the study. Condense by 50% as it adds little.

Decision letter (RSOS-201733.R0)

Dear Mr Jagielski,

The Editors assigned to your paper RSOS-201733 "Polar bears fail to forage optimally with declining resource density in a large seabird colony" have made a decision based on their reading of the paper and any comments received from reviewers.

Regrettably, in view of the reports received, the manuscript has been rejected in its current form. However, a new manuscript may be submitted which takes into consideration these comments.

We invite you to respond to the comments supplied below and prepare a resubmission of your manuscript. Below the referees' and Editors' comments (where applicable) we provide additional requirements. We provide guidance below to help you prepare your revision.

Please note that resubmitting your manuscript does not guarantee eventual acceptance, and we do not generally allow multiple rounds of revision and resubmission, so we urge you to make every effort to fully address all of the comments at this stage. If deemed necessary by the Editors, your manuscript will be sent back to one or more of the original reviewers for assessment. If the original reviewers are not available, we may invite new reviewers.

Please resubmit your revised manuscript and required files (see below) no later than 02-May-2021. Note: the ScholarOne system will 'lock' if resubmission is attempted on or after this deadline. If you do not think you will be able to meet this deadline, please contact the editorial office immediately.

Please note article processing charges apply to papers accepted for publication in Royal Society Open Science (<https://royalsocietypublishing.org/rsos/charges>). Charges will also apply to papers transferred to the journal from other Royal Society Publishing journals, as well as papers submitted as part of our collaboration with the Royal Society of Chemistry

(<https://royalsocietypublishing.org/rsos/chemistry>). Fee waivers are available but must be requested when you submit your manuscript
(<https://royalsocietypublishing.org/rsos/waivers>).

Thank you for submitting your manuscript to Royal Society Open Science and we look forward to receiving your resubmission. If you have any questions at all, please do not hesitate to get in touch.

on behalf of Professor Leslie Brown (Associate Editor) and Pete Smith (Subject Editor)
openscience@royalsociety.org

Associate Editor Comments to Author (Professor Leslie Brown):

We thank you for submitting the manuscript to our journal. The study is worthwhile and provides important information on these very important animals. However, both reviewers are of the opinion that the manuscript will need to be refocused with some extra analyses or at least the stating of the constraints of the analyses due to the small sample size and short time of survey. The dataset is regarded as too small to make generalized statements on optimal foraging theory. Furthermore there are many contentious statements made without enough substantiation from either the data or literature. Based on the large amount of comments/queries from the reviewers it would be needed that the manuscript is revised and refocused substantially addressing all the comments of the reviewers and resubmitted for review. Please pay careful attention to all the comments and suggestions made by the reviewers. We hope that you would consider resubmitting it to the journal.

Reviewer comments to Author:

Reviewer: 1

Comments to the Author(s)

This study used drones to examine the foraging behavior of polar bears at common eider nests on Mitvik Island during a 10 day period in 2017. The authors focused on determining whether behavior follows patterns expected per optimal foraging theory. The authors provide some interesting and informative results about polar bear behavior at seabird colonies. This is particularly important because direct observational studies of polar bear foraging are overall quite rare, including of their more traditional prey on the sea ice. Thus, despite the small sample size and short duration of the study, the data are important to publish. That said, more caution seems warranted in generalizing the potential results. At most the data are from 20 bears and we don't know how few that might be (3, 5, 10?). Thus, I'd suggest that the authors be somewhat more cautious in the conclusions drawn. I struggled to link the analyses to the objectives. I had to read it several times to make sense of what was done and why, and didn't fully understand what some of the analyses were getting at until I read the discussion (i.e. the models were clear but the rationale was not). More background is needed to support that the foraging bout order is an appropriate proxy for resource abundance (see specific comments below) since that is a key aspect of the paper (and the terms need to be used consistently - foraging bout versus event, etc.). Importantly, I think more caution is warranted when discussing the context of polar bear use of bird eggs broadly (see my specific comments below) as it is overstated in several places. The conclusion of the paper is that polar bears are not yet efficient predators of seabird eggs, yet I

didn't find the evidence for that conclusion convincing, in part because of the low sample size and short duration of the study but also based on the metrics examined (i.e. frequency of turning and selectivity). I would be more convinced that their "behavior" wasn't optimized which is different than saying they aren't efficient predators - the latter really requires knowing whether they balanced energetic gain with the expense of foraging. I think that some of the most important findings here are not highlighted. For example, it is significant that you documented a decline in nest visitation and increase in encountering of empty nests in just a 10 day period by less than 20 bears. This suggests that it is a quickly depleted resource that is not likely to sustain bears for any appreciable amount of time unless within that 10 days they are packing on extra pounds and there are similar sites throughout this region? - not to mention long-term sustainability of the food resources as a result of potential decline in eider populations via reduced reproduction. Is Mitivik an anomaly or are there similar seabird nesting throughout this region that other polar bears may be using (i.e., can some context be given for whether this site is similar to many others in the region or is somewhat rare?). Discussing these topics would make this a greater and more significant contribution than it is currently. I look forward to seeing a revised, published version.

Specific comments

Abstract: Lines 15-17: The statement "In response [to climate mediated sea-ice loss] polar bears are increasingly supplementing ice-based seal-hunting opportunities with eggs from seabird colonies". It's important to note that although there have been reports of polar bears feeding on eggs from seabird colonies (both in the past and in recent years), that few show this increasing over time and associated with sea ice loss (other than that they overlap more because bears come onshore earlier). Overall, this is not a common behavior by polar bears (i.e. there are not a lot of bears exhibiting this behavior even if some bears in a number of locations exhibit this behavior). The vast majority of bears that come to land in the summer throughout the polar bears range either rest and wait for the sea ice to return or scavenge on marine mammal carcasses. Thus, this statement is overgeneralized and should be edited to better reflect the context of observed consumption of seabird eggs by polar bears.

Line 34: It's not clear why the word "continually" is used in this sentence.

Line 72: This sentence sets up the expectation that you are measuring intake rates. I'd suggest rewording this section to indicate that you examined whether bears are responsive to declining resource density (which was assumed to occur over time during the observations) by modifying their selectivity, visiting fewer empty nests, etc.

Line 74: Can you be more specific for #2. How might it be expected that they would adjust their movement behaviors in relation to resource density?

Line 75: What would lower quality items be if all the of the resources are eggs? Eggs at various stages and nutritional content?

More detail needs to be provided on the expected directionality and rationale for the relationships examined (i.e. that you would expect them to become less selective as the resource is depleted, etc.)

Materials & methods

Study site: It would be helpful to reference Fig. 2 here and further describe the habitat used by common eiders (i.e. they are nesting in ground nests among vegetation and rocky areas, etc.).

Information on the timeframe in which they create nests and lay eggs as well as how synchronous egg-laying is are important context for the study.

Line 113: Is a foraging event the same as a foraging bout?

Line 117-119: Using this information it would be helpful to have some idea of the possible minimum number of bears (i.e. could this be repeated data from 3 bears or is it more likely 10 or more?). This is important because the manuscript is based on a very temporally constrained data set (i.e. 10 days of data in a single year).

Line 122-124: This isn't entirely convincing since experience would likely affect questions #1 and #4 at a minimum, if not all of the questions identified in the study (which you acknowledge later in the discussion). Perhaps here you could at least note potential effects of individual experience (either within the year of your study or between years) on behavior.

Line 140-144: It is really interesting that this behavior even occurs. I would have assumed they would eat eggs in any nest. Additional background on bear behavior at the nests would be helpful when you present the objectives.

Lines 157-158: "This analysis is meant to contextualize as the season progresses..." Here you need to go ahead and explain how you added the temporal component in the analysis. You explain it in the statistical analyses, but I was left wondering about how that was being incorporated and didn't get the answer until the statistical analyses section. Also, it's helpful earlier on to understand that you used foraging bout/event

Line 170-172: I think I've answered my question here that foraging event is the same as foraging bout? Table 1 indicated "foraging event". I'm reviewing the manuscript assuming these are the same thing. Obviously, if that is the case, the same term needs to be used throughout. Also, in line 172, perhaps clarify that foraging bout/event is a proxy for trends or changing resource density in addition to indicating that it was used as a continuous variable.

Statistical analyses:

Lines 167-172: I struggled to relate the models to the objectives. It would be easier to understand the analysis if Table 1 included a column indicating the question/objective being addressed with each model and had a separate column for the response/dependent variable and independent model variables. The table heading needs to indicate what the foraging event, search time and number of cues used are including that they are continuous variables, even though you have stated that information in the text. This allows the table to stand alone to help the reader understand the variety of models.

At this stage in the paper, I'm still unclear how the models address the questions of "were bears able to limit search time and maximize intake by...." stated in the introduction. I understand that temporal change is assumed to be a result of resource density. But I'm not sure what the expectation is relative to movement sinuosity or selectivity over time (with foraging event).

Should they become more selective and move less b/c the return is less? Similarly what is the expectation for movement sinuosity relative to search time? Should they use visual cues more as foraging event increases (i.e. resource density increases). In the models, I'd suggest using a different term than foraging event that you define that would make the models more intuitively related to the questions (i.e. perhaps you use "resource density" as the model parameter but state in the text and in tables & figures that the a proxy of chronological foraging event was used).

It's also difficult for the reader to understand and assess how well foraging event might serve as a proxy for resource density. This seems like it would be the case if all bears are foraging in the

same locations such that the order of bouts would definitely affect resource density. But this is unclear from the background provided. Further, how synchronous is egg-laying? Does that play a role in the chronological sequence of resource density?

Results: Line 196 “as predicted..”. I don’t remember seeing predictions. As stated previously, it has been unclear up to this point what was expected and why some of the relationships were examined.

Line 200-203: I’m not following the logic here. If they exhibited a trend of increasing visits to empty nests how does that demonstrate the ability to discern when nests are empty? Visiting fewer nests overall makes sense. Also, the relationship with visiting empty nests was significant at $p = 0.08$. You should be specific any time you discuss a result that has a $p > 0.05$ as most readers assume that relationships identified in a discussion are significant at $p \leq 0.05$

Figs. 3 and 4 use the term “foraging bout order” – similar comment as previously, I understand this to be the same as a foraging event? Perhaps use “foraging event order” throughout?

Lines 221-223: There is so little background provided on why the number of turns is indicative of optimal foraging that it’s impossible to evaluate this finding. In the introduction the expected relationship needs to be described (which I presume is more sinuosity when there are more nests). The information in lines 233-236 should be provided in the introduction (i.e. this was the hypothesis you were testing).

Lines 236-237: I again question the potential effects of the assumption that foraging event/bout/bout order (which I understand to all be the same measure) is perfectly measuring resource abundance. Either you need to make a stronger case for that or acknowledge the potential that your proxy is imperfect and could have affected the lack of an apparent relationship with sinuosity.

Lines 245-257: Any thoughts on the basis for selectivity? What is known about nutritional quality of the eggs as they incubate? Do other predators avoid eggs with fecal matter as you suggest? If you have no idea what might cause selectivity, even saying that would be informative. It is surprising to me that they are selective.

Line 280: Perhaps reword so “not necessarily” is not included twice in the same sentence.

Line 300-302: This statement is not convincing. Assessing whether polar bears can “adapt to a lengthier terrestrial residency” is not fully addressed by determining whether they use foraging strategies similarly in terrestrial and sea ice environments and/or exhibit behavior consistent with optimal foraging theory. Ultimately it depends on the quality and abundance of food (and their intake of it) in relation to the energetic cost to obtain it.

Line 302-304: Your results don’t provide any information about maximum net energy gain. The conclusions you can make are only about behaviors relative to optimal foraging theory.

Line 307: satiated from what? Having just come off the ice and feeding on seals? If so, is there evidence of recent feeding (i.e., defecating) that might indicate whether bears are feeding just prior to come to the island?

Line 308-310: This seems pretty speculative given the data (i.e. collected over a 10 day period in a single year). In the next sentence you acknowledge the small sample size, but I’d suggest not trying to link the short-term behavioral observations in this study to potential implications for body condition.

Reviewer: 2

Comments to the Author(s)

This is an interesting study on the foraging ecology of an obligate carnivore on an unusual prey item. The sample size is small and I urge the authors greater caution in interpretation. The current manuscript is overly speculative and a more tentative approach would be helpful. Nonetheless, it is a novel study and quite a challenging data set to obtain.

I urge greater caution in presentation of what is known about the foraging ecology of polar bears and its importance. There is heavy reliance on works by Gormezano et al. yet those studies are similar in their findings to Russell (1975 - Arctic 28: 117-129). It is unclear that terrestrial resources use is increasing or that it has any significant energetic value (see Rode et al. 2015).

I have concerns about the conclusion of non-optimal foraging. I think, correct me if I'm incorrect, but you have a staggered entry design for bears. The "trial", in an optimal foraging sense, restarts for every new individual. Because you are not following an individual through time and optimal foraging is an individual response, I don't see the matching between theory and the study design. I'm open to being convinced otherwise.

The paper is well written. I have some specific issues but with some careful reconsideration, the manuscript will make an interesting contribution to the literature.

16 - "increasingly supplementing ice-based seal-hunting" - there is limited support for this statement. I agree that it is increasingly reported but I suspect that this is related, at least in part, to the amount of activity in the Arctic and interest in polar bears. Some of the very earliest reports of polar bears from explorers record eating of eggs (e.g., Samuel Hearne, John Rae). Possibly "increased reports of supplemental feeding" or something similar.

44 - the statement "poses any long-term negative consequences to species impacted" precludes any positive consequences. This seems like an odd statement and somewhat biased.

53-54 - "increasing the bears' reliance on alternative prey" - you cite a book and 3 papers by a single group of authors. The idea that there is increasing reliance on alternative prey, however, Rode KD, Robbins CT, Nelson L, Amstrup SC. 2015. Can polar bears use terrestrial foods to offset lost ice-based hunting opportunities? *Frontiers in Ecology and the Environment* 13:138-145.

This paper states "Only small numbers of polar bears have been documented consuming terrestrial foods even in modest quantities" and "Where consumption of terrestrial foods has been documented, polar bear body condition and survival rates have declined even as land use has increased. Thus far, observed consumption of terrestrial food by polar bears has been insufficient to offset lost ice-based hunting opportunities"

A better reflection of the state of knowledge on terrestrial feeding by polar bears is warranted.

57 - a peer reviewed source would be preferable to a report

57 - please clarify what is meant by "This" - is it referring to "conserve" or "foraging performance"

59 - I agree that the cues are different but there is clear evidence in the literature that foraging on birds is not a new behavior and goes far beyond the noted references.

65 – I find the argument that polar bears may have poor foraging performance a bit odd. Polar bears evolved from a brown bear ancestry (fairly recently) and brown bears are omnivores with a very diverse diet. As an olfactory predatory, why should polar bears have poor foraging performance? Polar bears are known to consume a wide variety of species despite their specialization so postulating good performance seems as likely as it being poor. Authors choice on this but it strikes me that a stronger approach is that seaduck eggs may be a novel forage item and thus the situation is akin to prey switching in functional response context.

79-81 – “To our knowledge, this is the first study...” I’m not a fan of such justification – the science should speak for itself. I suggest removing this as it add little.

96-113 – the study is based on 10 days of sampling in a single year, 995 minutes of sampling, a maximum of 20 bears, and 31 foraging events. Given the challenges in collecting the data, this is a small but reasonable sample size. However, it does restrict the ability to generalize.

121 – “new individual” – I have a sense that there is a likelihood of pseudoreplication and that some animals could have been counted repeatedly as being new. This further restricts the works ability to generalize.

124 – optimal foraging theory is an interesting context for the study but it is unclear how patches may be assessed in this context. A polar bear may view the whole island as a patch and have limited ability to move. Within the context of an ideal free distribution, the bears may not be free to move (i.e., swim to a new patch) and know of alternatives. The interactions with other bears may not be free. Because polar bears foraging on terrestrial prey lose mass, I’m not convinced that optimal foraging is the driving force / context. I’m OK with it but do the bears have a choice to move? Is the marginal value theorem operating only at the scale of the island and not the broader landscape? Will a depleted patch (the island) result in a bear moving to a new patch?

142 – I think the argument that feces covered eggs are not consumed? This seems very odd – polar bears are notorious for eating decaying carrion. That duck feces is a deterrent seems a bit odd to me. I’ve not seen evidence of similar behaviour in other egg eating mammals.

Methods – there is no information on how the nests are located / mapped. A brief section on this would be useful. I did not see this in the supplementary files.

172 – watching the videos, I imagine the depletion of nests is also due to gulls? Arctic foxes too?

184-5 – “with bears marginally increasing their visits to empty nests” – this is where low statistical power is hampering analyses. 1 year of data with only 31 observations. There’s no solution but it warrants a caveat rich discussion.

I don’t see how individual was assessed in the analyses. I assume some of the observations are repeated for some individuals and others have a single sample? If this is a cross-section of animals, the implications / conclusions may be quite different. For example, a bear on its 10 foraging trip may behave quite differently from a bear on its first trip. I don’t see a path to addressing this but again, caution in interpretation is warranted.

193-195 – I wasn’t a fan of “first” in the introduction but getting a double dose is even less appealing. Being first is not a meaningful reason for publication.

195 – I don’t see how a bear consuming an egg cannot gain energy. Reword / clarify.

196-197 – I have some concerns about the conclusion (and thus the title) because you are not following individuals over time. This is a cross-sectional study and you presumably have a staggered entry into the “trial”. I don’t see a means of getting past this but assuming non-optimal foraging appears to be a major overreaching of the data / design.

201-207 – yes, animals can detect food. This is in essence the argument being made. A horse can see that there is no grass to be eaten. A crow will scavenge from a carcass it can see. I would reduce this emphasis / condense. It’s not an overly novel finding.

214 – the genus is now *Pusa* for *Phoca hispida*

214 – no information is provided on ice conditions. To this point, I did not understand that the bears had the choice to hunt for seals on the ice vs. be on land. More detail on sea ice conditions is needed. Do you have information on how long bears stayed on the island? Did some leave? If the context is seaduck eggs vs. seals, this has to be made clear. If it is this case, eating eggs may yield benefits beyond energy (e.g., micronutrients, vitamins).

229 – “foraging in avian colonies appear to forego the consumption of many nests in their direct foraging path” – to me, this observation suggests that eggs are not a meaningful source of energy. If this is true, the idea of optimal foraging is not meaningful. A core tenant of optimal foraging is that the animal must be foraging. This suggests that the animal is not feeding on available resources, thus not depleting them, and thus, unlikely to forage optimally.

238 – “satiation” – given that a polar bear can consume ca. 10-20% of its body mass and a bear could weigh 500 kg. Conservatively, satiation might be 50 kg of eggs and at 150 grams/egg (based on a goose egg mass), satiation might come with 333 eggs (plus / minus). If there are 1600 pairs with a mean clutch of 4 eggs (possibly high), this means there are ca. 6400 eggs available pre-predation. Does it make sense to invoke a bear eating 5% of the eggs (333/6400) in a single bout? With 31 bouts observed, does this make sense? Invoking satiation is OK but there’s no information on the bears provided. If the bears are small (say 200 kg), satiation may make more sense. Your citation on lines 287-291 suggest that my calculations are conservative. The alternate hypothesis, not proposed, is that they’re not that motivated / interested in feeding. It is noteworthy that fasting ursids often do not feed even when presented with food. Did 100% of the bears observed feed? It is odd that fasting physiology was not explored as an aspect of a bear’s motivation to forage (or not). Perhaps the authors are implying satiation is the equivalent of fasting?

255-257 – delete – this is akin to “do more research” and add little

283-286 – “we therefore postulate polar bears can potentially learn to optimize (and potentially specialize [77, 78]) their foraging performance should they continually be driven onto nearshore terrestrial environments” This is a bit odd given the analyses of Rode et al. (op. cit.) that suggests terrestrial resources are not a meaningful contribution. If eider eggs were a meaningful energy source, should bears exploit them already? In an ideal free context, shouldn’t they give up hunting seals to forage on eggs even if sea ice is present? Maybe my logic is off but if the eggs are a meaningful source of energy, they should be used.

Another caveat, I suspect that heavy predation on an eider colony would reduce their reuse, predictability for the bears, and cause a decline in eiders using the area. The longer term dynamics seem tenuous.

For this section, I urge less speculation.

310 – the caveat of a small sample size shows up very late in the discussion. Move to the start. Readers need a reminder of this from the outset.

296-314 – this is largely a summary / review of the study. Condense by 50% as it adds little.

===PREPARING YOUR MANUSCRIPT===

- one version identifying all the changes that have been made (for instance, in coloured highlight, in bold text, or tracked changes);
- a 'clean' version of the new manuscript that incorporates the changes made, but does not highlight them.

This version will be used for typesetting if your manuscript is accepted.

===PREPARING YOUR REVISION IN SCHOLARONE===

- 1) One version identifying all the changes that have been made (for instance, in coloured highlight, in bold text, or tracked changes);
 - 2) A 'clean' version of the new manuscript that incorporates the changes made, but does not highlight them.
 - An individual file of each figure (EPS or print-quality PDF preferred [either format should be produced directly from original creation package], or original software format).
 - An editable file of each table (.doc, .docx, .xls, .xlsx, or .csv).
 - An editable file of all figure and table captions.
- Note: you may upload the figure, table, and caption files in a single Zip folder.
- Any electronic supplementary material (ESM).
 - If you are requesting a discretionary waiver for the article processing charge, the waiver form must be included at this step.
 - If you are providing image files for potential cover images, please upload these at this step, and inform the editorial office you have done so. You must hold the copyright to any image provided.
 - A copy of your point-by-point response to referees and Editors. This will expedite the preparation of your proof.

- Ensure that your data access statement meets the requirements at <https://royalsociety.org/journals/authors/author-guidelines/#data>. You should ensure that you cite the dataset in your reference list. If you have deposited data etc in the Dryad repository, please include both the 'For publication' link and 'For review' link at this stage.
- If you are requesting an article processing charge waiver, you must select the relevant waiver option (if requesting a discretionary waiver, the form should have been uploaded at Step 3 'File upload' above).
- If you have uploaded ESM files, please ensure you follow the guidance at <https://royalsociety.org/journals/authors/author-guidelines/#supplementary-material> to include a suitable title and informative caption. An example of appropriate titling and captioning may be found at https://figshare.com/articles/Table_S2_from_Is_there_a_trade-off_between_peak_performance_and_performance_breadth_across_temperatures_for_aerobic_sc_ope_in_teleost_fishes_/3843624.

Author's Response to Decision Letter for (RSOS-201733.R0)

See Appendix A.

Decision letter (RSOS-210391.R0)

Dear Mr Jagielski,

I am pleased to inform you that your manuscript entitled "Polar bears are inefficient predators of seabird eggs" is now accepted for publication in Royal Society Open Science.

on behalf of Professor Leslie Brown (Associate Editor) and Pete Smith (Subject Editor)
openscience@royalsociety.org

Associate Editor Comments to Author (Professor Leslie Brown):
Associate Editor

Comments to the Author:

Thank you for the changes affected and the detailed explanations given. The manuscript in my opinion is now acceptable for publication.

Appendix A

Associate Editor Comments to Author (Professor Leslie Brown):

We thank you for submitting the manuscript to our journal. The study is worthwhile and provides important information on these very important animals. However, both reviewers are of the opinion that the manuscript will need to be refocused with some extra analyses or at least the stating of the constraints of the analyses due to the small sample size and short time of survey. The dataset is regarded as too small to make generalized statements on optimal foraging theory. Furthermore there are many contentious statements made without enough substantiation from either the data or literature. Based on the large amount of comments/queries from the reviewers it would be needed that the manuscript is revised and refocused substantially addressing all the comments of the reviewers and resubmitted for review. Please pay careful attention to all the comments and suggestions made by the reviewers. We hope that you would consider resubmitting it to the journal.

Reviewer comments to Author:

Reviewer: 1

Comments to the Author(s)

This study used drones to examine the foraging behavior of polar bears at common eider nests on Mitivik Island during a 10 day period in 2017. The authors focused on determining whether behavior follows patterns expected per optimal foraging theory. The authors provide some interesting and informative results about polar bear behavior at seabird colonies. This is particularly important because direct observational studies of polar bear foraging are overall quite rare, including of their more traditional prey on the sea ice. Thus, despite the small sample size and short duration of the study, the data are important to publish. That said, more caution seems warranted in generalizing the potential results. At most the data are from 20 bears and we don't know how few that might be (3, 5, 10?). Thus, I'd suggest that the authors be somewhat more cautious in the conclusions drawn. I struggled to link the analyses to the objectives. I had to read it several times to make sense of what was done and why, and didn't fully understand what some of the analyses were getting at until I read the discussion (i.e. the models were clear but the rationale was not). More background is needed to support that the foraging bout order is an appropriate proxy for resource abundance (see specific comments below) since that is a key aspect of the paper (and the terms need to be used consistently – foraging bout versus event, etc.). Importantly, I think more caution is warranted when discussing the context of polar bear use of bird eggs broadly (see my specific comments below) as it is overstated in several places. The conclusion of the paper is that polar bears are not yet efficient predators of seabird eggs, yet I didn't find the evidence for that conclusion convincing, in part because of the low sample size and short duration of the study but also based on the metrics examined (i.e. frequency of turning and selectivity). I would be more convinced that their “behavior” wasn't optimized which is different than saying they aren't efficient predators – the latter really requires knowing whether they balanced energetic gain with the expense of foraging. I think that some of the most important findings here are not highlighted. For example, it is significant that you documented a decline in nest visitation and increase in encountering of empty nests in just a 10 day period by less than

20 bears. This suggests that it is a quickly depleted resource that is not likely to sustain bears for any appreciable amount of time unless within that 10 days they are packing on extra pounds and there are similar sites throughout this region? - not to mention long-term sustainability of the food resources as a result of potential decline in eider populations via reduced reproduction. Is Mitivik an anomaly or are there similar seabird nesting throughout this region that other polar bears may be using (i.e., can some context be given for whether this site is similar to many others in the region or is somewhat rare?). Discussing these topics would make this a greater and more significant contribution than it is currently. I look forward to seeing a revised, published version.

Thank you for your comments. You are correct in saying that our sample size is small and the conclusions we drew were maybe overarching with respect to the presented data. We took your specific comments/recommendations to reconcile this issue and drew more reasonable conclusions represented by the data (see below for specific changes).

Specific comments

1) Abstract: Lines 15-17: The statement “In response [to climate mediated sea-ice loss] polar bears are increasingly supplementing ice-based seal-hunting opportunities with eggs from seabird colonies”. It’s important to note that although there have been reports of polar bears feeding on eggs from seabird colonies (both in the past and in recent years), that few show this increasing over time and associated with sea ice loss (other than that they overlap more because bears come onshore earlier). Overall, this is not a common behavior by polar bears (i.e. there are not a lot of bears exhibiting this behavior even if some bears in a number of locations exhibit this behavior). The vast majority of bears that come to land in the summer throughout the polar bears range either rest and wait for the sea ice to return or scavenge on marine mammal carcasses. Thus, this statement is overgeneralized and should be edited to better reflect the context of observed consumption of seabird eggs by polar bears.

Thank you for pointing this out. We changed it to: (Lines 11-13)

“Climate-mediated sea-ice loss is disrupting the foraging ecology of polar bears (Ursus maritimus) across much of their range. As a result, there have been increased reports of polar bears foraging on seabird eggs across parts of their range.”

2) Line 34: It’s not clear why the word “continually” is used in this sentence.

There was no particular reason, so we removed it for clarity.

3) Line 72: This sentence sets up the expectation that you are measuring intake rates. I’d suggest rewording this section to indicate that you examined whether bears are responsive to declining resource density (which was assumed to occur over time during the observations) by modifying their selectivity, visiting fewer empty nests, etc.

Agreed! Thank you. We changed it to: (Lines 75-78)

“Therefore, we examined whether foraging bears are responsive to declining resource density by sampling the area and exhibiting behaviours in accordance with expectations of OFT that would minimize net energetic expenditure.”

4) Line 74: Can you be more specific for #2. How might it be expected that they would adjust their movement behaviors in relation to resource density?

Yes, we added: (Lines 81-85 & 93-94)

“(2) Adjusting their movement sinuosity in relation to resource density (i.e., area-restricted search theory [54, 55]) as a means of limiting energetically costly time spent searching and maximizing nest encounter rates (i.e., slow and sinuous movement in a concentrated area when resources are in high abundance versus faster walking speed and more straight-line movement when resources become less abundant).”

And

“(2) move with greater sinuosity early in the season when full clutches are more abundant versus more straight-line energy-minimizing movement as clutches deplete”

5) Line 75: What would lower quality items be if all of the resources are eggs? Eggs at various stages and nutritional content?

By lower quality items, we are referring to eggs that are soiled with eider faeces, which is one of their anti-predator adaptations (McDougall P, Milne H. 1978. The anti-predator function of defecation on their own eggs by female eiders. *Wildfowl*. 29). The faeces of nesting eiders is more alkaline than non-nesting eiders, so we hypothesised that this may irritate the polar bears senses.

6) More detail needs to be provided on the expected directionality and rationale for the relationships examined (i.e. that you would expect them to become less selective as the resource is depleted, etc.)

Thank you for the feedback. We added details and adjusted this section as follows: (Lines 73-98)

“In a previous study in this system, polar bears were observed to consume clutches at a decelerating rate (indicating declining resource density) until they depleted the colony of eggs [49]. Therefore, we examined whether foraging bears are responsive to declining resource density by sampling the area and exhibiting behaviours in accordance with expectations of OFT that would minimize net energetic expenditure. Specifically, by: (1) using nest site information, such as odour of a clutch or conspicuousness of eggs [50], to avoid ‘already predated’ nests [51] since foraging in an already-searched-area is both time and energetically costly and may have implications on patch-residency time decisions (i.e., Marginal Value Theorem [52, 53]). (2) Adjusting their movement sinuosity in relation to resource density (i.e., area-restricted search theory [54, 55]) as a means of limiting energetically costly time spent searching and maximizing nest encounter rates (i.e., slow and sinuous movement in a

concentrated area when resources are in high abundance versus faster walking speed and more straight-line movement when resources become less abundant). (3) Modifying their ‘selectivity’ in ingesting clutches of eggs of perceived ‘lower quality’ (i.e., eggs covered in eider faeces, which is the result of the eiders’ predatory defence mechanism [56], as the highly alkaline faecal matter may irritate predators) in accordance with resource availability (i.e., foragers are more selective when resources are in high abundance versus less selective as the resource depletes [57-59]). We also examined (4) whether using visual cues (i.e., flushing eider hens) aids bears to locate nests [60-62]. We predicted that as the season progressed and resources declined, bears arriving later in the season would demonstrate different behaviours to those arriving early. Specifically, bears would: (1) visit significantly fewer empty nests later in the season even as fewer nests are visited overall; (2) move with greater sinuosity early in the season when full clutches are more abundant versus more straight-line energy-minimizing movement as clutches deplete; (3) be more choosy (i.e., ignore more clutches) early in the season when full clutches are abundant. We also predicted that (4) bears would use visual cues to locate nests throughout the season regardless of resource abundance. Given that terrestrial resources are becoming increasingly used by polar bears [27], our goal was to determine if polar bears foraging on seabird eggs behave in a manner consistent with the expectations of OFT.”

Materials & methods

7) Study site: It would be helpful to reference Fig. 2 here and further describe the habitat used by common eiders (i.e. they are nesting in ground nests among vegetation and rocky areas, etc.). Information on the timeframe in which they create nests and lay eggs as well as how synchronous egg-laying is are important context for the study.

Agreed!

By the time this study took place (i.e., July 10th -20th) most eiders would have laid their eggs 2 weeks prior. See Schematic below!

We added this section in the methods for clarification: (Lines 110-116)

“Birds nest in ‘cups’ on the ground within the mossy and rocky terrain of the island (Figure 2). In 2017 (our study year), eiders arrived to the island approximately in late June (mean arrival (julian) date: 172.12 ± 0.22 days (June 21st); median arrival date: 172.25 ± 0.31 days (June 21st)); and initiated laying a few days later (mean lay date: 175.65 ± 0.60 days (June 25th); median lay date: 175.71 ± 0.83 days (June 25th)). Our observations took place on July 10 to 20, which is approximately midway through this colony’s mean incubation stage [64], so nest replenishment (due to relaying post-predation) or late lay initiation was likely minimal during this study.”

Figure. Timeline schematic showing: a) eider mean laying date, b) mean mid-incubation date, c) mean hatch date, d) dates when eider researchers discouraged bears from coming onto the island (2017), e) to f) dates when bears were filmed (2017), g) date eider colony depleted (2017). Note: colony depletion occurred prior to mean hatch date. Data for a-c from Mitivik Island (2002-2008) (Love et al. 2010, *Oecologia* 164: 277-286)

8) Line 113: Is a foraging event the same as a foraging bout?

Yes, “foraging bout” and “foraging event” is the same thing. We should have caught that earlier on. Thank you. We changed it to “foraging bout” throughout for consistency.

9) Line 117-119: Using this information it would be helpful to have some idea of the possible minimum number of bears (i.e. could this be repeated data from 3 bears or is it more likely 10 or more?). This is important because the manuscript is based on a very temporally constrained data set (i.e. 10 days of data in a single year).

In short, based on field notes, we can confidentially say that there were at least 17 bears on the Island during this study period (although it is unknown how many of those were foraging for consecutive days). Based on experience, most bears will stay around the island for one day, but in rare cases a few days. Differentiating bears was a challenging task since the drone flew at a relatively high altitude to minimize disturbance. However, whenever a bear had an obvious distinguishing mark within a time period of less than 15-20 minutes (give or take), we were confident enough to stitch those files together and consider it a “foraging bout”. Sometimes, that same individual would show up later in the day to forage, hence more than one foraging bout in a day (see table in supplementary material). Of all

the foraging bouts, 7 bears had more than one foraging bout but only 3 bears had up to three foraging bouts. If we were not confident in the identification of a particular individual (i.e., the markings were not visible in the particular footage), we erred on the side of caution and considered it a different bear, acknowledging that it could in fact be the same individual from earlier. Because bears are so difficult to differentiate, we considered bears to be new individuals across days (barring identifiable features, size differences). We still believe that by studying the effects of a depleting resource over days, the variability of the main effect (i.e., time) would swamp out any variation from individual-level effects.

We added: (Lines 148 - 155)

“We acknowledge identifying individual bears within and across days was a challenge and strove to find identifying features when possible. However, because we observed multiple bears on a given day (C.J. Dey and E.S. Richardson, pers. obs.), we can assert any random variance from individual-level effects across days would be swamped by the main effect of the declining resource over time. In addition, while OFT is generally used to describe the resource use of an individual, this framework remains applicable to our study since OFT can act as a guide to describe general foraging behaviours at the population-, guild-, and species levels without the tracking of individual ID [67-69], with such methods also being applied to Ursids [59].”

10) Line 122-124: This isn't entirely convincing since experience would likely affect questions#1 and #4 at a minimum, if not all of the questions identified in the study (which you acknowledge later in the discussion). Perhaps here you could at least note potential effects of individual experience (either within the year of your study or between years) on behavior.

Excellent point! Thank you for bringing this up. While we did add those points initially, we ended up rewording the text in lines 148-155, as noted above, (and included much more information on OFT throughout) and feel that it no longer fits in this section. But please see how we refocused the section in the discussion pertaining to this very point (Lines 313-323).

“While our results suggest that polar bears do not necessarily adhere to all expectations of OFT, this is not surprising given that lengthier terrestrial residency has only recently become an increasing occurrence for polar bears at a population level [31, 32]. Variation in foraging efficiency may be due to differences in bears' experience levels, and polar bear terrestrial foraging behaviours may simply reflect a period of reinforced learning [82, 83]. We therefore postulate that polar bears can improve their foraging performance should they continually be driven onto nearshore terrestrial environments and overlap with the incubation period of ground nesting birds [as seen here: 34, 39-43]. Given that polar bears are capable of learning to use resources that have proven to be profitable in the past [84], individuals who have experienced foraging success in avian colonies, such as those reported consuming hundreds of waterfowl eggs over a few days [39, 44], may be driven via memory to forage on eggs over several years and, as a result, become more adept at the task.”

11) Line 140-144: It is really interesting that this behavior even occurs. I would have assumed they would eat eggs in any nest. Additional background on bear behavior at the nests would be helpful when you present the objectives.

Agreed, it is odd, isn't it? There is not much more to say than: *“(i.e., looked into the nest, sniffed the clutch) and then ignored it....”* However, we added: (Lines 194) *“For further detail on behaviours at a nest, see [49].”*

12) Lines 157-158: *“This analysis is meant to contextualize as the season progresses...”* Here you need to go ahead and explain how you added the temporal component in the analysis. You explain it in the statistical analyses, but I was left wondering about how that was being incorporated and didn't get the answer until the statistical analyses section. Also, it's helpful earlier on to understand that you used foraging bout/event

We added the following paragraphs earlier on in the text to clarify: (Lines 73-75 and 140-142)

“In a previous study in this system, polar bears were observed to consume clutches at a decelerating rate (indicating declining resource density) until they depleted the colony of eggs [49].”

And

“Importantly, each video was timestamped and foraging bouts were placed in chronological order which served as a proxy for resource depletion (i.e., declining resource density over time) as bears consumed clutches at a decelerating rate until they depleted the colony [49].”

13) Line 170-172: I think I've answered my question here that foraging event is the same as foraging bout? Table 1 indicated “foraging event”. I'm reviewing the manuscript assuming these are the same thing. Obviously, if that is the case, the same term needs to be used throughout. Also, in line 172, perhaps clarify that foraging bout/event is a proxy for trends or changing resource density in addition to indicating that it was used as a continuous variable.

Yes, foraging bout and foraging event are the same. We changed everything in the text to “foraging bout” for consistency

Also, see response to Reviewer One # 12

Additionally, we reiterate this in the Statistical analyses section: (Lines 204-207)

“For models a-d, we used foraging bout order as a continuous predictor variable (i.e., first recorded foraging bout = 1, last recorded foraging bout = 31) which serves as a proxy for resource density (since bears' clutch consumption rates decline with time [49] and are therefore indicative of a declining resource base).”

Statistical analyses:

14) Lines 167-172: I struggled to relate the models to the objectives. It would be easier to understand the analysis if Table 1 included a column indicating the question/objective being addressed with each model and had a separate column for the response/dependent variable and independent model variables.

The table heading needs to indicate what the foraging event, search time and number of cues used are including that they are continuous variables, even though you have stated that information in the text. This allows the table to stand alone to help the reader understand the variety of models.

Thank you for the suggestion. We included a new column in Table 1 indicating the objectives. Also, here is the new table caption:

“Statistical models (a-e) used to test drivers of polar bear foraging behaviours on common eider eggs at Mitivik Island, Nunavut. For models a-d, foraging bout order is the continuous predictor variable (i.e., first recorded foraging bout = 1, last recorded foraging bout = 31) as it serves as a proxy for resource density since bears continually consumed clutches until depleting the colony [49]. For model e, proportion of cues used (i.e., cues used divided by cues observed by bears) is the predictor variable, since it encompasses the entire suite of events when a cue(s) was present and available for a bear to use. For models a-e, searching time length (i.e., time spent walking and standing during a foraging bout) is added as a fixed effect to account for differences in filmed-video lengths.”

15) At this stage in the paper, I’m still unclear how the models address the questions of “were bears able to limit search time and maximize intake by....” stated in the introduction. I understand that temporal change is assumed to be a result of resource density. But I’m not sure what the expectation is relative to movement sinuosity or selectivity over time (with foraging event). Should they become more selective and move less b/c the return is less? Similarly what is the expectation for movement sinuosity relative to search time? Should they use visual cues more as foraging event increases (i.e. resource density increases). In the models, I’d suggest using a different term than foraging event that you define that would make the models more intuitively related to the questions (i.e. perhaps you use “resource density” as the model parameter but state in the text and in tables & figures that the a proxy of chronological foraging event was used).

Thank you for the suggestions. See responses to Reviewer One #s 6, 12 and 13

In our Tables, we also added “resource density” as you suggested and “search duration” for clarification. We defined searching time (194-196), explained why we used search duration as a fixed effect (209-211) and reiterated this in our Table 1 caption.

16) It’s also difficult for the reader to understand and assess how well foraging event might serve as a proxy for resource density. This seems like it would be the case if all bears are foraging in the same locations such that the order of bouts would definitely affect resource

density. But this is unclear from the background provided. Further, how synchronous is egg-laying? Does that play a role in the chronological sequence of resource density?

See response to Reviewer One # 12 and 7, respectively.

17) Results: Line 196 “as predicted..”. I don’t remember seeing predictions. As stated previously, it has been unclear up to this point what was expected and why some of the relationships were examined.

Predictions are now clearly stated (Lines 90-96).

18) Line 200-203: I’m not following the logic here. If they exhibited a trend of increasing visits to empty nests how does that demonstrate the ability to discern when nests are empty? Visiting fewer nests overall makes sense. Also, the relationship with visiting empty nests was significant at $p = 0.08$. You should be specific any time you discuss a result that has a $p > 0.05$ as most readers assume that relationships identified in a discussion are significant at $p \leq 0.05$

Thank you, we acknowledged that our result is marginally significant at the 0.1 level in the Results (Line 219) and modified the Discussion section: (Lines 236-252)

“As the eider breeding season advanced, total visits of bears to nests (i.e., clutches eaten, empty nests, and nests ignored) declined (Figure 3a). Simultaneously, bears showed a marginal increase in their visits to empty nests (contrary to predictions; Figure 2a; 3b), suggesting that the proportion of ‘nest visits’ are increasingly empty which may indicate that bears are unable to discern full- from already predated nests in advance of visiting them [51], either by using sensory mechanisms such as odour of a clutch or conspicuousness of eggs [50]. Given that visiting empty nests can be both time- and energetically costly (i.e., by visiting already searched grounds), the seeming inability to recognize which nests are empty before visiting them may have important implications on optimal giving-up-times/densities [52, 53] in that bears may overstay in the colony. While the energetic cost of foraging in the Mitivik Island eider colony is low [49] and likely has minimal fitness implications as bears walk slowly across a flat terrain and dense colony, we posit that an optimal giving-up-time would become particularly critical for bears foraging either late in the eider-breeding season when few full nests are left, or in a larger area with fewer nesting hens. Quantifying an optimal give-up-time of polar bears foraging on eggs at a larger spatio-temporal extent would aid in determining polar bears’ overall ability to make adaptive patch-residency time decisions (i.e., Marginal Value Theorem [52, 53]). Although this is well beyond the scope of the present study, our findings present the first step into a more thorough investigation of long-term fitness consequences of terrestrial foraging for polar bear populations.”

19) Figs. 3 and 4 use the term “foraging bout order” – similar comment as previously, I understand this to be the same as a foraging event? Perhaps use “foraging event order” throughout?

We changed the text from “foraging event” to “foraging bout” to align with the figures which still say ‘foraging bout order.’ We also added in Lines 139-142:

“Importantly, each video was timestamped and foraging bouts were placed in chronological order which served as a proxy for resource depletion (i.e., declining resource density over time) as bears consumed clutches at a decelerating rate until they depleted the colony [49].”

20) Lines 221-223: There is so little background provided on why the number of turns is indicative of optimal foraging that it's impossible to evaluate this finding. In the introduction the expected relationship needs to be described (which I presume is more sinuosity when there are more nests). The information in lines 233-236 should be provided in the introduction (i.e. this was the hypothesis you were testing).

Yes, we added more details in the introduction. See response to Reviewer One # 4 and 6

21) Lines 236-237: I again question the potential effects of the assumption that foraging event/bout/bout order (which I understand to all be the same measure) is perfectly measuring resource abundance. Either you need to make a stronger case for that or acknowledge the potential that your proxy is imperfect and could have affected the lack of an apparent relationship with sinuosity.

See response to Reviewer One # 12

22) Lines 245-257: Any thoughts on the basis for selectivity? What is known about nutritional quality of the eggs as they incubate? Do other predators avoid eggs with fecal matter as you suggest? If you have no idea what might cause selectivity, even saying that would be informative. It is surprising to me that they are selective.

Yes, some predators do avoid eggs soiled in fecal matter (McDougall P, Milne H. 1978 The anti-predator function of defecation on their own eggs by female eiders. Wildfowl. 29) The faeces of nesting eiders is more alkaline than non-nesting eiders, so we hypothesised that this may irritate the polar bears senses. The nutritional content of eggs changes through time (more nutritional early in the incubation period), so if anything, bears should be more selective as the season progresses (i.e., less nutrition in the eggs).

23) Line 280: Perhaps reword so “not necessarily” is not included twice in the same sentence.

Thank you for the suggestion. We changed it to: (Lines 313-315)

“While our results suggest that polar bears do not necessarily adhere to all expectations of OFT, this is not surprising given that lengthier terrestrial residency has only recently become an increasing occurrence for polar bears at a population level [31, 32].”

24) Line 300-302: This statement is not convincing. Assessing whether polar bears can “adapt to a lengthier terrestrial residency” is not fully addressed by determining whether they use foraging strategies similarly in terrestrial and sea ice environments and/or exhibit behavior consistent with optimal foraging theory. Ultimately it depends on the quality and abundance of food (and their intake of it) in relation to the energetic cost to obtain it.

Thank you for your comment. We completely restructured the manuscript throughout: See response to Reviewer Two #9.

25) Line 302-304: Your results don't provide any information about maximum net energy gain. The conclusions you can make are only about behaviors relative to optimal foraging theory.

We changed it to: (Lines 338-334)

“Our findings here suggest that polar bears are inefficient predators of seabird eggs, particularly in the context of matching foraging behaviours to resource density. If polar bears are limited in their ability to accurately assess the quality of their foraging patch and do not adjust their behaviours accordingly, the energy individual bears gain from eggs may be less than previously estimated [49]. These results may therefore influence the previously predicted energetic consequences of this climate-mediated behavioural shift at a population-level [89].”

26) Line 307: satiated from what? Having just come off the ice and feeding on seals? If so, is there evidence of recent feeding (i.e., defecating) that might indicate whether bears are feeding just prior to come to the island?

There was no evidence of feeding on seals (i.e., scat), no. But given that bears are continually migrating on the sea ice towards Mitivik Island to follow more Northern ice-floes for feeding, it is likely their movement is foraging-based, and therefore satiation is a distinct possibility.

Satiation was removed from this particular paragraph in the Conclusion, but we clarified this more in the Discussion: (Lines 272-277 and 304-305):

“i) their inexperience foraging in areas of high prey density in a seabird colony given that their typical marine prey occurs at low densities and are encountered infrequently [38], and/or ii) satiation from recent feeding in the marine environment prior to coming onto the island as part of their foraging-based migration (and therefore no need to employ a search mode). However, we observed bears actively searching for nests on the island late in the season despite that few clutches remained, making satiation an unlikely universal explanation.”

And

“...and/or iii) bears were satiated (again, possibly from recently feeding on the ice) with no need to actively seek out cued-nests.”

27) Line 308-310: This seems pretty speculative given the data (i.e. collected over a 10 day period in a single year). In the next sentence you acknowledge the small sample size, but I'd suggest not trying to link the short-term behavioral observations in this study to potential implications for body condition.

See response to Reviewer One # 25

Reviewer: 2

Comments to the Author(s)

This is an interesting study on the foraging ecology of an obligate carnivore on an unusual prey item. The sample size is small and I urge the authors greater caution in interpretation. The current manuscript is overly speculative and a more tentative approach would be helpful. Nonetheless, it is a novel study and quite a challenging data set to obtain.

I urge greater caution in presentation of what is known about the foraging ecology of polar bears and its importance. There is heavy reliance on works by Gormezano et al. yet those studies are similar in their findings to Russell (1975 – Arctic 28: 117-129). It is unclear that terrestrial resources use is increasing or that it has any significant energetic value (see Rode et al. 2015).

I have concerns about the conclusion of non-optimal foraging. I think, correct me if I'm incorrect, but you have a staggered entry design for bears. The "trial", in an optimal foraging sense, restarts for every new individual. Because you are not following an individual through time and optimal foraging is an individual response, I don't see the matching between theory and the study design. I'm open to being convinced otherwise.

The paper is well written. I have some specific issues but with some careful reconsideration, the manuscript will make an interesting contribution to the literature.

Thank you for your comments. We agree that our sample size is small (but useable!) and we therefore adjusted our conclusions to better match the data. We also added a substantial amount of background in the Introduction and Discussion on polar bear terrestrial foraging ecology.

Specifically, we clarified the text regarding polar bears use of terrestrial resources (see Reviewer Two #1, 3 and 6), and how optimal foraging theory is an appropriate framework to use for this study (see Reviewer Two #16)

1) 16 – “increasingly supplementing ice-based seal-hunting” - there is limited support for this statement. I agree that it is increasingly reported but I suspect that this is related, at least in part, to the amount of activity in the Arctic and interest in polar bears. Some of the very earliest reports of polar bears from explorers record eating of eggs (e.g., Samuel Hearne, John Rae). Possibly “increased reports of supplemental feeding” or something similar.

Thank you for pointing this out. We changed it to: (Lines 11-13):

“Climate-mediated sea-ice loss is disrupting the foraging ecology of polar bears (*Ursus maritimus*) across much of their range. As a result, there have been increased reports of polar bears foraging on seabird eggs across parts of their range.”

On a side note: We point to Smith et al. 2010. Polar Biology & Iverson et al. 2014. Proc. Royal Soc. Biol. Sci., and anecdotal evidence from H.G. Gilchrist and O.P. Love who have seen first-hand over nearly a quarter century a stark increase in polar bear predation on eider eggs (to the point where we have not seen eider duckling survival in the last ~ 5 seasons) at our long-standing research site.

2) 44 – the statement “poses any long-term negative consequences to species impacted” precludes any positive consequences. This seems like an odd statement and somewhat biased.

Thank you for this point. We have adjusted it to encompass the possibility of adaptation (rather than just mentioning consequences) which hopefully makes the statement less biased: (Lines 39-43)

“The question of whether species can adapt to procuring alternative prey efficiently (and thus maximize net energetic gains), or whether foraging on alternative prey poses long-term negative consequences (or simply reflects a period of adjustment), first requires exploring whether foraging performance, and by extension, foraging decisions, follow predictions of OFT.”

3) 53-54 – “increasing the bears’ “reliance on alternative prey” – you cite a book and 3 papers by a single group of authors. The idea that there is increasing reliance on alternative prey, however, Rode KD, Robbins CT, Nelson L, Amstrup SC. 2015. Can polar bears use terrestrial foods to offset lost ice-based hunting opportunities? *Frontiers in Ecology and the Environment* 13:138-145.

This paper states “Only small numbers of polar bears have been documented consuming terrestrial foods even in modest quantities” and “Where consumption of terrestrial foods has been documented, polar bear body condition and survival rates have declined even as land use has increased. Thus far, observed consumption of terrestrial food by polar bears has been insufficient to offset lost ice-based hunting opportunities”

A better reflection of the state of knowledge on terrestrial feeding by polar bears is warranted.

Thank you for this point! We made changes throughout the manuscript: (Lines 49-55)

“Although polar bears traditionally feed on seals and other marine mammals hunted on the sea-ice [23, 24], they are known to opportunistically forage on land as well, with reports of such behaviours dating back to the 15th century [25]. In recent years, however, the inclusion of terrestrial resources has apparently increased for polar bears occupying the southernmost extent of their range [26, 27;], potentially as a result of missed ice-based hunting opportunities due to changes in sea ice phenology [21]. Some authors have suggested that terrestrial resources may help offset lost ice-based hunting opportunities [28, 29], although this assertion remains unknown [30].”

(Lines 324-336)

“4.6 Long term profitability and sustainability of this predator-prey interaction are unknown

Recent years have seen an increase in reports of polar bears foraging in eider colonies [42-44]. However, most of these reports (including this study) note only a handful of bears in the colonies and, even with over 300 000 breeding common eider hens inhabiting the Canadian Arctic [85], only approximately 30% of colonies are visited by bears on a regional scale [42]. Polar bears also deplete the resource quickly and only a few individuals gain appreciable calories [49]. Combined, these lines of evidence suggest eider eggs are not yet a significant resource for polar bears at a population level [30], so while efficient foraging behaviour may impact the number of calories an individual bear accrues, the overall importance of eggs as a resource for polar bears will remain low unless bears' use of available colonies intensifies. However, should the number of bears foraging in eider colonies increase, the consequences for eider populations may be devastating [86, 87], making this resource unsustainable in the long run [but see 88 where predictive modelling suggests otherwise under certain conditions].”

4) 57 – a peer reviewed source would be preferable to a report

Agreed, updated with this article:

[Ref # 36]. Vongraven, D., Aars, J., Amstrup, S., Atkinson, S. N., Belikov, S., Born, E. W., ... & Lunn, N. (2012). A circumpolar monitoring framework for polar bears. *Ursus*, 23(sp2), 1-66.

5) 57 – please clarify what is meant by “This” – is it referring to “conserve” or “foraging performance”

Thank you for the comment. It was meant to say “foraging performance”, which we changed it to.

6) 59 – I agree that the cues are different but there is clear evidence in the literature that foraging on birds is not a new behavior and goes far beyond the noted references.

Agreed that this is not a new behaviour observed in polar bears, but this was not the intent of our argument. The argument made here is that a greater number of bears are reportedly making greater use of this resource than in the past. We wished to study the decision-making behaviours of bears specifically with respect to egg foraging – a still relatively uncommon alternative resource - and whether and how they sampled their environment. At our research site, we have noted this increase in predation rate on clutches over a 20-year period (Iverson et al. 2014; and given the co-authors working at Mitivik for a quarter of a century and observing this first-hand). While polar bears have been documented to forage in avian colonies in the past, this was only a few individuals, so this had no real significant meaning to the population overall. However, the pattern of more and more bears using this resource is increasing (albeit still not at a population level), making this a biologically and ecologically relevant phenomenon.

7) 65 – I find the argument that polar bears may have poor foraging performance a bit odd. Polar bears evolved from a brown bear ancestry (fairly recently) and brown bears are omnivores with a very diverse diet. As an olfactory predatory, why should polar bears have poor foraging performance? Polar bears are known to consume a wide variety of species despite their specialization so postulating good performance seems as likely as it being poor. Authors choice on this but it strikes me that a stronger approach is that seaduck eggs may be a novel forage item and thus the situation is akin to prey switching in functional response context.

Thank you for raising the point.

We want to clarify that it is not their foraging ability, but rather their foraging behaviours that are not “optimal”. The idea that bears should be good foragers because they are related to brown bears makes sense. However, we think that is also fair that they would have poor foraging performance based on the fact that polar bears evolved to hunt marine prey that occur at low densities (Stirling I. 1974 Can. J. Zool. 52, 1191-1198; i.e., have no need to adjust behaviours to density).

8) 79-81 – “To our knowledge, this is the first study...” I’m not a fan of such justification – the science should speak for itself. I suggest removing this as it add little.

Agreed. We removed it!

9) 96-113 – the study is based on 10 days of sampling in a single year, 995 minutes of sampling, a maximum of 20 bears, and 31 foraging events. Given the challenges in collecting the data, this is a small but reasonable sample size. However, it does restrict the ability to generalize.

Agreed! We have tried to cut back on generalizing throughout the manuscript based on reviewer comments.

We acknowledged that the results from our study only represent a small percentage of bears within a population (229-234), that the consequences of poor foraging performance won’t necessarily impact body condition (338-343), and that this resource may not be sustainable in the long run (326-336).

However, we still feel strongly our results are worthwhile to report in light of the increasing (and relatively new) phenomenon of bears foraging on avian eggs to a greater extent than in the past. We have already quantified the energetic consequences of this behaviour (Jagielski et al. 2021. Animal Behaviour) and think that there is much to learn from an evolutionary context (i.e., bears foraging performance on a relatively new resource) as well.

“We acknowledge that our sample size is small (i.e., based on one season of observations) and that our results need to be taken in context of island size and eider-nest density. However, based on this present study, our results suggest that polar bear foraging performance does not

align with expectations of Optimal Foraging Theory (OFT), in that bears do not behave in accordance to resource density and as a result, do not exhibit evidence of foraging efficiency.”

“Our findings here suggest that polar bears are inefficient predators of seabird eggs, particularly in the context of matching foraging behaviours to resource density. If polar bears are limited in their ability to accurately assess the quality of their foraging patch and do not adjust their behaviours accordingly, the energy individual bears gain from eggs may be less than previously estimated [49]. These results may therefore influence the previously predicted energetic consequences of this climate-mediated behavioural shift at a population-level [89].

“Recent years have seen an increase in reports of polar bears foraging in eider colonies [42-44]. However, most of these reports (including this study) note only a handful of bears in the colonies and, even with over 300 000 breeding common eider hens inhabiting the Canadian Arctic [85], only approximately 30% of colonies are visited by bears on a regional scale [42]. Polar bears also deplete the resource quickly and only a few individuals gain appreciable calories [49]. Combined, these lines of evidence suggest eider eggs are not yet a significant resource for polar bears at a population level [30], so while efficient foraging behaviour may impact the number of calories an individual bear accrues, the overall importance of eggs as a resource for polar bears will remain low unless bears’ use of available colonies intensifies. However, should the number of bears foraging in eider colonies increase, the consequences for eider populations may be devastating [86, 87], making this resource unsustainable in the long run [but see 88 where predictive modelling suggests otherwise under certain conditions].”

10) 121 – “new individual” – I have a sense that there is a likelihood of pseudoreplication and that some animals could have been counted repeatedly as being new. This further restricts the works ability to generalize.

See response to Reviewer One #9

11) 124 – optimal foraging theory is an interesting context for the study but it is unclear how patches may be assessed in this context. A polar bear may view the whole island as a patch and have limited ability to move. Within the context of an ideal free distribution, the bears may not be free to move (i.e., swim to a new patch) and know of alternatives. The interactions with other bears may not be free. Because polar bears foraging on terrestrial prey lose mass, I’m not convinced that optimal foraging is the driving force / context. I’m OK with it but do the bears have a choice to move? Is the marginal value theorem operating only at the scale of the island and not the broader landscape? Will a depleted patch (the island) result in a bear moving to a new patch?

We treated the island as a single patch bears would forage in, and tested whether they could assess patch quality and exhibit behaviours accordingly. Polar bears in our video footage did not seem (for the most part) to focus in particular areas exclusively on the island, at least not micro-areas. Bears would walk around, seemingly haphazardly sometimes. Also, it is challenging to delineate arbitrary patches (Arditi, R., & Dacorogna, B. (1988). Optimal foraging on arbitrary food distributions and the definition of habitat patches. *The American Naturalist*, 131(6), 837-846); and given the physical characteristics

of the island (flat and small: 24hectares) and nesting distribution of the ducks (scattered around most of the island), we decided that all reasons listed contribute collectively to our decision to delineate the island as one patch. We agree that such delineation would become extremely important at a larger landscape scale, however, and that bears will move on to a new patch.

12) 142 – I think the argument that feces covered eggs are not consumed? This seems very odd – polar bears are notorious for eating decaying carrion. That duck feces is a deterrent seems a bit odd to me. I've not seen evidence of similar behaviour in other egg eating mammals.

Some predators do avoid eggs soiled in fecal matter (McDougall P, Milne H. 1978 The anti-predator function of defecation on their own eggs by female eiders. *Wildfowl*. 29). The faeces of nesting eiders is more alkaline than non-nesting eiders, so we hypothesised that this may irritate the polar bears senses. The fact that bears ignore any nests at all is a curiosity in and of itself (as noted too, by Reviewer One).

13) Methods – there is no information on how the nests are located / mapped. A brief section on this would be useful. I did not see this in the supplementary files.

Nests on the island are not mapped. We were able to visibly discern from the videos nests as being empty vs. full. As foraging events happened chronologically, and eiders nest relatively synchronously, we are making the assumption that successful foraging results in resource depletion.

14) 172 – watching the videos, I imagine the depletion of nests is also due to gulls? Arctic foxes too?

While Arctic foxes do appear on the island from time to time, this occurs only early in the breeding season (June) and discouraged by researchers and thus, they likely do not impact eggs depletion to a great extent. There is likely some additive eggs loss from the herring gulls on the island, and is a research question our team is currently pursuing.

15) 184-5 – “with bears marginally increasing their visits to empty nests” – this is where low statistical power is hampering analyses. 1 year of data with only 31 observations. There's no solution but it warrants a caveat rich discussion.

Thank you for the point, we stated in the Results that this increase visits to empty nests was marginally significant at the 0.1 level (Line 219). We also shored up the Discussion section (Lines 236 -252)

“As the eider breeding season advanced, total visits of bears to nests (i.e., clutches eaten, empty nests, and nests ignored) declined (Figure 3a). Simultaneously, bears showed a marginal increase in their visits to empty nests (contrary to predictions; Figure 2a; 3b), suggesting that the proportion of ‘nest visits’ are increasingly empty which may indicate that bears are unable to discern full- from already predated nests in advance of visiting them [51],

either by using sensory mechanisms such as odour of a clutch or conspicuousness of eggs [50]. Given that visiting empty nests can be both time- and energetically costly (i.e., by visiting already searched grounds), the seeming inability to recognize which nests are empty before visiting them may have important implications on optimal giving-up-times/densities [52, 53] in that bears may overstay in the colony. While the energetic cost of foraging in the Mitivik Island eider colony is low [49] and likely has minimal fitness implications as bears walk slowly across a flat terrain and dense colony, we posit that an optimal giving-up-time would become particularly critical for bears foraging either late in the eider-breeding season when few full nests are left, or in a larger area with fewer nesting hens. Quantifying an optimal give-up-time of polar bears foraging on eggs at a larger spatio-temporal extent would aid in determining polar bears' overall ability to make adaptive patch-residency time decisions (i.e., Marginal Value Theorem [52, 53]). Although this is well beyond the scope of the present study, our findings present the first step into a more thorough investigation of long-term fitness consequences of terrestrial foraging for polar bear populations.”

16) I don't see how individual was assessed in the analyses. I assume some of the observations are repeated for some individuals and others have a single sample? If this is a cross-section of animals, the implications / conclusions may be quite different. For example, a bear on its 10 foraging trip may behave quite differently from a bear on its first trip. I don't see a path to addressing this but again, caution in interpretation is warranted.

Some bears were repeat foragers, yes, but none more than 3 times. See response to Reviewer One #9

OFT acts as a guide for bears being able to assess their patch quality (given that they're closely related to grizzlies and should therefore have retained these terrestrial-foraging adaptations); and therefore, bears, regardless if repeat or new, should be able to integrate information of a resource in low abundance/availability, and behaviour accordingly.

In a similar study looking at brown bear 'choosiness' behaviour, the authors did not track individuals, but still used OFT as their guiding framework (Lincoln and Quinn, 2019. Behavioural Ecology).

We added: (Lines 148-155)

“We acknowledge identifying individual bears within and across days was a challenge and strove to find identifying features when possible. However, because we observed multiple bears on a given day (C.J. Dey and E.S. Richardson, pers. obs.), we can assert any random variance from individual-level effects across days would be swamped by the main effect of the declining resource over time. In addition, while OFT is generally used to describe the resource use of an individual, this framework remains applicable to our study since OFT can act as a guide to describe general foraging behaviours at the population-, guild-, and species levels without the tracking of individual ID [67-69], with such methods also being applied to Ursids [59].”

17) 193-195 – I wasn't a fan of “first” in the introduction but getting a double dose is even less appealing. Being first is not a meaningful reason for publication.

Agreed! Removed.

18) 195 – I don't see how a bear consuming an egg cannot gain energy. **Reword / clarify.**

“some” referred to the fact that at the end of the season, when eggs were depleted, some bears were unable to gain energy as they did not find any eggs. For simplicity, we took out the word “some”.

19) 196-197 – I have some concerns about the conclusion (and thus the title) because you are not following individuals over time. This is a cross-sectional study and you presumably have a staggered entry into the “trial”. I don't see a means of getting past this but assuming non-optimal foraging appears to be a major overreaching of the data / design.

Thanks for the point. We took it into consideration and dampened our conclusions throughout (see Discussion), including a new title: *Polar bears are inefficient predators of seabird eggs*. See also our response to Reviewer Two #16.

20) 201-207 – yes, animals can detect food. This is in essence the argument being made. A horse can see that there is no grass to be eaten. A crow will scavenge from a carcass it can see. I would reduce this emphasis / condense. It's not an overly novel finding.

See response to # 15 directly above!

21) 214 – the genus is now *Pusa* for *Phoca hispida*

Thank you! Changed.

22) 214 – no information is provided on ice conditions. To this point, I did not understand that the bears had the choice to hunt for seals on the ice vs. be on land. More detail on sea ice conditions is needed. Do you have information on how long bears stayed on the island? Did some leave? If the context is seaduck eggs vs. seals, this has to be made clear. If it is this case, eating eggs may yield benefits beyond energy (e.g., micronutrients, vitamins).

In response to your comment above and overall comments on speculation, we took out this section as we do not have sea-ice data and agree that this statement was over speculative.

23) 229 – “foraging in avian colonies appear to forego the consumption of many nests in their direct foraging path” – to me, this observation suggests that eggs are not a meaningful source of energy. If this is true, the idea of optimal foraging is not meaningful. A core tenant of optimal foraging is that the animal must be foraging. This suggests that the animal is not feeding on available resources, thus not depleting them, and thus, unlikely to forage optimally.

A few things. All bears in this study foraged. That is, all bears were observed searching and consuming eggs except for the bears on the very last days as the colony was fully depleted (Jagielski et al. 2021. Animal Behaviour). However, the bears on the last days were walking

around the island and some were even observed trying to reach something (eggs, dead eider?) in a crevasse. Polar bears would walk by several nests early in the season (i.e., when the colony was full – but still consume eggs, which provide an estimated 2803kj/clutch) and fewer nests as the season went on and resource depleted. So, we would even argue that given the fact that they were foraging, and the fact that they forewent consuming many nests (when the colony was full), they were in fact not engaging in area-restricted searching (or at least not as well as they could have).

24) 238 – “satiation” – given that a polar bear can consume ca. 10-20% of its body mass and a bear could weigh 500 kg. Conservatively, satiation might be 50 kg of eggs and at 150 grams/egg (based on a goose egg mass), satiation might come with 333 eggs (plus / minus). If there are 1600 pairs with a mean clutch of 4 eggs (possibly high), this means there are ca. 6400 eggs available pre-predation. Does it make sense to invoke a bear eating 5% of the eggs (333/6400) in a single bout? With 31 bouts observed, does this make sense? Invoking satiation is OK but there’s no information on the bears provided. If the bears are small (say 200 kg), satiation may make more sense. Your citation on lines 287-291 suggest that my calculations are conservative. The alternate hypothesis, not proposed, is that they’re not that motivated / interested in feeding. It is noteworthy that fasting ursids often do not feed even when presented with food. Did 100% of the bears observed feed? It is odd that fasting physiology was not explored as an aspect of a bear’s motivation to forage (or not). Perhaps the authors are implying satiation is the equivalent of fasting?

This is a fantastic observation/point. You are correct. A bear would have to eat a lot of eggs to become satiated, and we did not record those very high numbers per foraging bout. We also do not have actual sizes of the bears, so that does not help. The point being made (and we made it clearer in the text), is that a bear might have just recently consumed a seal. Also, while a lack of feeding motivation is a valid point, we strongly believe that that was not the case for bears in this study as all animals foraged, or at least tried to feed (see response above).

25) 255-257 – delete – this is akin to “do more research” and add little

Deleted!

26) 283-286 – “we therefore postulate polar bears can potentially learn to optimize (and potentially specialize [77, 78]) their foraging performance should they continually be driven onto nearshore terrestrial environments” This is a bit odd given the analyses of Rode et al. (op. cit.) that suggests terrestrial resources are not a meaningful contribution. If eider eggs were a meaningful energy source, should bears exploit them already? In an ideal free context, shouldn’t they give up hunting seals to forage on eggs even if sea ice is present? Maybe my logic is off but if the eggs are a meaningful source of energy, they should be used.

While animals are capable of specializing on a resource as a result of being excluded from their main resource (Araújo MS, Bolnick DI, Layman CA. 2011 The ecological causes of individual specialisation. *Ecol. Lett.* 14, 948-958), we took this part out of the text to reduce speculation.

Seals are much more calorie dense than eggs so if sea ice is available, bears should continue hunting seals (although see Pagano et al. 2020 *Ecology*. 101, e02959). However, for the few individuals who do forage in avian colonies, over time, they will likely be able to improve their foraging performance as they gain experience.

We changed the section to: (Lines 313-323)

“While our results suggest that polar bears do not necessarily adhere to all expectations of OFT, this is not surprising given that lengthier terrestrial residency has only recently become an increasing occurrence for polar bears at a population level [31, 32]. Variation in foraging efficiency may be due to differences in bears’ experience levels, and polar bear terrestrial foraging behaviours may simply reflect a period of reinforced learning [82, 83]. We therefore postulate that polar bears can improve their foraging performance should they continually be driven onto nearshore terrestrial environments and overlap with the incubation period of ground nesting birds [as seen here: 34, 39-43]. Given that polar bears are capable of learning to use resources that have proven to be profitable in the past [84], individuals who have experienced foraging success in avian colonies, such as those reported consuming hundreds of waterfowl eggs over a few days [39, 44], may be driven via memory to forage on eggs over several years and, as a result, become more adept at the task.”

27) Another caveat, I suspect that heavy predation on an eider colony would reduce their reuse, predictability for the bears, and cause a decline in eiders using the area. The longer term dynamics seem tenuous.

Great Point! See response to Reviewer Two # 3

For this section, I urge less speculation.

We did our best to reduce any speculation throughout the Discussion!

28) 310 – the caveat of a small sample size shows up very late in the discussion. Move to the start. Readers need a reminder of this from the outset.

Agreed. We added this to our first paragraph in the discussion: (Lines 229-234)

“We acknowledge that our sample size is small (i.e., based on one season of observations) and that our results need to be taken in context of island size and eider-nest density. However, based on this present study, our results suggest that polar bear foraging performance does not align with expectations of Optimal Foraging Theory (OFT), in that bears do not behave in accordance to resource density and as a result, do not exhibit evidence of foraging efficiency.”

29) 296-314 – this is largely a summary / review of the study. Condense by 50% as it adds little.

Agreed, we changed it to: (Lines 338-348)

“Our findings here suggest that polar bears are inefficient predators of seabird eggs, particularly in the context of matching foraging behaviours to resource density. If polar bears are limited in their ability to accurately assess the quality of their foraging patch and do not adjust their behaviours accordingly, the energy individual bears gain from eggs may be less than previously estimated [49]. These results may therefore influence the previously predicted energetic consequences of this climate-mediated behavioural shift at a population-level [89]. We acknowledge that our sample size is small, and therefore suggest that a long-term spatial and temporal study is required to better understand not only the long-term fitness consequences of polar bear foraging performance beyond Mitivik Island, but also the ecological implications of this predator-prey interaction. This study demonstrates that, while species are able to incorporate “less preferred” resources into their diet when their primary prey becomes more difficult to obtain, they may not be able to do so efficiently.”